# Precocious expression of Blimp1 in B cells causes autoimmune disease with increased self-reactive plasma cells

Peter Bönelt[1], Miriam Wöhner[1,†], Martina Minnich[1,†], Hiromi Tagoh[1,‡], Maria Fischer[1], Markus Jaritz[1], Anoop Kavirayani[2], Manasa Garimella[3], Mikael CI Karlsson[3] & Meinrad Busslinger[1,*] iD

## Abstract

The transcription factor Blimp1 is not only an essential regulator of plasma cells, but also a risk factor for the development of autoimmune disease in humans. Here, we demonstrate in the mouse that the *Prdm1* (Blimp1) gene was partially activated at the chromatin and transcription level in early B cell development, although mature *Prdm1* mRNA did not accumulate due to post-transcriptional regulation. By analyzing a mouse model that facilitated ectopic Blimp1 protein expression throughout B lymphopoiesis, we could demonstrate that Blimp1 impaired B cell development by interfering with the B cell gene expression program, while leading to an increased abundance of plasma cells by promoting premature plasmablast differentiation of immature and mature B cells. With progressing age, these mice developed an autoimmune disease characterized by the presence of autoantibodies and glomerulonephritis. Hence, these data identified ectopic Blimp1 expression as a novel mechanism, through which Blimp1 can act as a risk factor in the development of autoimmune disease.

**Keywords** autoimmune disease; Blimp1-mediated loss of B cells; ectopic expression throughout the B cell lineage; increased plasma cells differentiation; *Prdm1* (Blimp1) transcription in developing B cells
**Subject Categories** Immunology
**The EMBO Journal (2019) 38: e100010**

## Introduction

Plasma cells serve an important role in the acute response to infection and in long-term protection of the host by providing humoral immunity through continuous secretion of antibodies.

Moreover, plasma cells often contribute also to the pathogenesis of autoimmune disease by secreting self-reactive antibodies (Suurmond & Diamond, 2015; Tsokos *et al*, 2016). The zinc finger transcription factor Blimp1 (encoded by the *Prdm1* gene) is a key regulator of plasma cells (Nutt *et al*, 2007), which was initially discovered by its ability to induce plasmacytic differentiation upon ectopic expression in mature B cells (Turner *et al*, 1994). Within the B-lymphoid lineage, Blimp1 is predominantly expressed in antibody-secreting cells, where its highest expression is observed in quiescent long-lived plasma cells (Kallies *et al*, 2004). Consistent with this expression pattern, antibody-secreting cells are lost in mice with a B cell-specific deletion of the *Prdm1* gene, demonstrating that Blimp1 is essential for the generation of plasmablasts and plasma cells (Shapiro-Shelef *et al*, 2003; Kallies *et al*, 2007). Blimp1 expression is furthermore required in long-lived bone marrow plasma cells to maintain their secretory function (Tellier *et al*, 2016). Interestingly, the human *PRDM1* gene is frequently mutated on both alleles in activated B cell-like diffuse large B cell lymphoma (ABC-DLBCL), demonstrating that loss of the tumor-suppressor gene *PRDM1* contributes to lymphomagenesis by preventing plasma cell differentiation (Pasqualucci *et al*, 2006; Tam *et al*, 2006). In addition, genome-wide association studies (GWAS) have identified *PRDM1* as a susceptibility gene for the autoimmune diseases systemic lupus erythematosus (SLE) and rheumatoid arthritis (RA), as two informative single nucleotide polymorphisms (SNPs) have been specifically mapped in the intergenic region between the *PRDM1* and *ATG5* loci in SLE and RA patients (Gateva *et al*, 2009; Raychaudhuri *et al*, 2009; Zhou *et al*, 2011; Appendix Fig S1A). To date, there are, however, no functional data available that would causally implicate *PRDM1* in the pathogenesis of SLE or RA.

At the molecular level, Blimp1 functions as a transcriptional repressor and activator by recruiting chromatin-remodeling and histone-modifying complexes to its target genes in plasmablasts

---

1 Research Institute of Molecular Pathology (IMP), Vienna Biocenter (VBC), Vienna, Austria
2 Vienna Biocenter Core Facilities (VBCF), Vienna Biocenter (VBC), Vienna, Austria
3 Department of Microbiology, Tumor and Cell Biology, Karolinska Institute, Stockholm, Sweden
 *Corresponding author. Tel: +43 1 79730 3150; E-mail: busslinger@imp.ac.at
 †These authors contributed equally to this work
 ‡Present address: Ludwig Institute for Cancer Research, University of Oxford, Oxford, UK

(Minnich *et al*, 2016). Systematic analysis of regulated target genes identified multiple roles of Blimp1 in coordinating plasma cell differentiation (Minnich *et al*, 2016; Tellier *et al*, 2016). For instance, Blimp1 promotes the migration and adhesion of plasmablasts. It directly represses several key genes including those coding for the transcription factor Pax5, the co-activator CIITA, and the cytidine deaminase AID, which leads to the silencing of B cell-specific gene expression, antigen presentation and class switch recombination in plasmablasts, respectively (Shaffer *et al*, 2002; Minnich *et al*, 2016). Blimp1 furthermore directly activates genes, leading to increased expression of the plasma cell regulator IRF4 and proteins involved in immunoglobulin secretion (Minnich *et al*, 2016). Importantly, Blimp1 strongly induces the transcription of the immunoglobulin heavy-chain (*Igh*) and κ light-chain (*Igk*) genes and also regulates the posttranscriptional expression switch from the membrane-bound to secreted Ig heavy-chain protein in plasmablasts (Minnich *et al*, 2016; Tellier *et al*, 2016).

Little is so far known about how Blimp1 expression is regulated in plasmablasts and plasma cells. MicroRNAs (miR-30b,d,e and miR125b) and the RNA-binding protein ZFP36L1 have been implicated in the posttranscriptional regulation of Blimp1 expression by controlling mRNA decay and translation through binding to *Prdm1* 3′ UTR sequences in plasma cell lines (Gururajan *et al*, 2010; Nasir *et al*, 2012; Kassambara *et al*, 2017). At the transcriptional level, the *Prdm1* gene is known to be activated by the transcription factors IRF4, E2A, and E2-2 in plasma cells (Sciammas *et al*, 2006; Kwon *et al*, 2009; Gloury *et al*, 2016; Wöhner *et al*, 2016). Moreover, the regulatory landscape of the *Prdm1* locus is rather complex, as it consists of eight open chromatin regions that interact with the *Prdm1* promoter in plasmablasts and are located up to 272 kb upstream of the *Prdm1* gene (Wöhner *et al*, 2016).

Here, we demonstrate that the *Prdm1* locus is partially activated at the chromatin level already in early B cell development and is transcribed in the nucleus, although mRNA does not accumulate in the cytoplasm of B cells due to posttranscriptional regulation. To study the effect of ectopic Blimp1 expression in the B cell lineage, we generated a mouse model that resulted in early Blimp1 expression due to insertion of the Moloney murine leukemia virus (MoMLV) enhancer together with the IRES-*hCd2* (*ihCd2*) reporter gene between the stop codon and the 3′ UTR of the *Prdm1*^ihCd2 allele (Minnich *et al*, 2016). Consequently, *Prdm1* transcription was strongly activated, and the Blimp1 protein was expressed already from lymphoid progenitors throughout B cell development with highest expression being observed in pro-B, pre-B, and immature B cells in the bone marrow of *Prdm1*^ihCd2/ + mice. Blimp1 interfered with the normal B cell gene expression program by activating and repressing many genes, which led to decreased B cell development in *Prdm1*^ihCd2/ + mice. In contrast, plasma cell development was strongly increased in *Prdm1*^ihCd2/ + mice, consistent with the finding that immature and mature B cells had an enhanced potential to undergo *in vitro* plasmablast differentiation. With progressing age, *Prdm1*^ihCd2/ + mice developed an autoimmune phenotype characterized by the generation of anti-nuclear antibodies, immune complex deposition, and kidney pathology. Together, these data demonstrate that precocious Blimp1 expression can cause autoimmune disease.

# Results

## The *Prdm1* locus is transcriptionally active already in early B cell development

We previously characterized open chromatin regions (sites A-H) upstream of the *Prdm1* (Blimp1) gene in plasmablasts, which highly express Blimp1 (Wöhner *et al*, 2016; Fig 1A). To investigate the epigenetic status of the *Prdm1* locus in early B cell development, we mapped open chromatin regions and different histone modifications by ATAC-seq and ChIP-seq analyses in pro-B cells, which do not express Blimp1. Unexpectedly, the *Prdm1* promoter was partially open and contained bivalent chromatin as shown by the presence of active (H3K4me2, H3K4me3, H3K9ac) and repressive (H3K27me3) histone marks. The upstream regions A, B, and C were also partially activated in pro-B cells, as shown by the presence of bivalent chromatin at region A and a subset of open chromatin sites in regions B and C compared to plasmablasts (Fig 1A). Notably, the far upstream regions D, E, F, G, and H, which also interact with the *Prdm1* promoter in plasmablasts (Wöhner *et al*, 2016), were present in closed chromatin in pro-B cells in marked contrast to plasmablasts (Fig 1A). Together, these data indicate that the *Prdm1* locus has undergone partial epigenetic activation already in pro-B cells.

The observed activation at the chromatin level may indicate that the *Prdm1* gene is already transcribed during B cell development. To test this hypothesis, we measured the nascent transcript levels in bone marrow pro-B cells and splenic follicular B cells by global run-on sequencing (GRO-seq; Core *et al*, 2008). As shown in Fig 1B and C, *Prdm1* and its neighboring gene *Atg5* were similarly transcribed in bone marrow pro-B cells and splenic follicular (FO) B cells. Despite similar transcription rates, only a low amount of *Prdm1* mRNA could be detected by RNA-seq analysis in both cell types in contrast to the relatively high abundance of *Atg5* mRNA (Fig 1B and C). Hence, these data indicate that posttranscriptional regulation prevents the accumulation of *Prdm1* mRNA during B cell development.

Posttranscriptional control can be mediated by microRNAs that act principally through the control of mRNA decay and translation by binding to the 3′ untranslated region (3′ UTR) of mRNA (Pasquinelli, 2012). Alternatively, RNA-binding proteins interact with AU-rich elements (AREs) in the 3′ UTR, which promotes mRNA deadenylation and decay (Turner *et al*, 2014). Both mechanisms have been implicated in the posttranscriptional control of *Prdm1* mRNA (Gururajan *et al*, 2010; Nasir *et al*, 2012; Parlato *et al*, 2013; Kassambara *et al*, 2017). To investigate the role of the 3′ UTR in the posttranscriptional regulation of *Prdm1*, we deleted most (2,253 bp; 90%) of the 2,490-bp long 3′ UTR by CRISPR/Cas9-mediated mutagenesis, which left only one consensus ARE motif and one microRNA-binding site in the truncated 3′ UTR of the *Prdm1*^Δ3′U(90) allele (Appendix Fig S1B and C). Contrary to expectation, this large deletion did not lead to increased *Prdm1* mRNA accumulation (Appendix Fig S1D) or elevated Blimp1 protein expression (Appendix Fig S1E) in early B cells of *Prdm1*^Δ3′U(90)/Δ3′U(90) mice and had no effect on B cell development in these mice (Appendix Fig S1F). We conclude therefore that a large part (90%) of the 3′ UTR is dispensable for the posttranscriptional regulation of *Prdm1* expression.

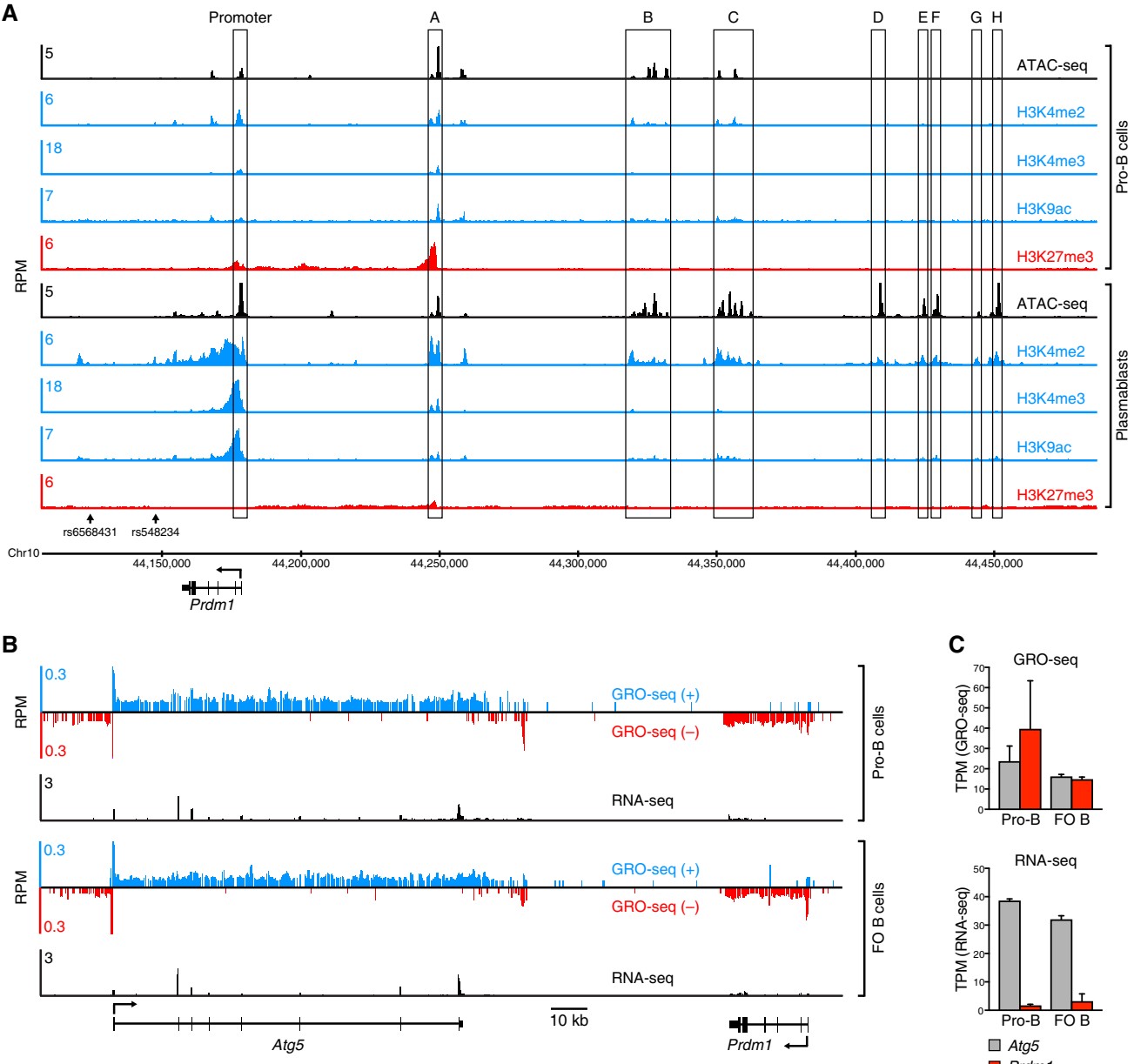

**Figure 1. Partial epigenetic and transcriptional activation of the *Prdm1* locus in B cells.**

A   Mapping of open chromatin as well as active and repressive histone modifications at upstream regulatory regions of the *Prdm1* locus in pro-B cells and plasmablasts. Open chromatin was determined by ATAC-seq (Buenrostro *et al*, 2013), and active (H3K4me2, H3K4me3, H3K9ac) and repressive (H3K27me3) histone marks were mapped by ChIP-seq analysis in *ex vivo* sorted pro-B cells (ATAC-seq and H3K27me3, this study), short-term *in vitro* cultured pro-B cells (H3K4me2, H3K4me3, H3K9ac; Revilla-i-Domingo *et al*, 2012), and *in vitro* LPS-induced plasmablasts (Minnich *et al*, 2016; Table EV4). The indicated upstream regions were previously shown by 3C analysis to interact with the *Prdm1* promoter (Wöhner *et al*, 2016). The mm9 genomic coordinates of mouse chromosome 10 and the respective positions of two human SNPs (rs658431 and rs548234) are indicated. RPM, reads per million mapped sequence reads.

B   Mapping of nascent transcripts or mature mRNA at the *Prdm1* and *Atg5* loci by GRO-seq or RNA-seq analyses of *ex vivo* sorted pro-B and FO B cells, respectively (Table EV4). Strand-specific GRO-seq reads are indicated in blue and red, respectively.

C   Quantification of nascent transcript and mRNA levels. The nascent transcription or mRNA expression of the *Atg5* and *Prdm1* genes is shown as mean expression value (TPM) with SEM based on two different GRO-seq or RNA-seq experiments for each B cell type, respectively. TPM, transcripts per million (Wagner *et al*, 2012).

## Premature expression of Blimp1 in lymphoid lineages of the *Prdm1*[ihCd2/+] mouse

Given the observed transcriptional activity of *Prdm1* in early B lymphopoiesis, we asked the question whether B cell development would be affected by precocious expression of the Blimp1 protein in the B cell lineage. To this end, we generated a mouse model facilitating ectopic Blimp1 expression throughout B lymphopoiesis by inserting a 258-bp long terminal repeat (LTR) sequence of the Moloney murine leukemia virus (MoMLV) together with the

IRES-hCd2 (ihCd2) reporter gene between the Prdm1 stop codon and the 3′ UTR (Fig 2A) in the Prdm1ihCd2 allele (Minnich et al, 2016). The insertion contains both copies of the 75-bp repeat of the MoMLV enhancer, which binds multiple transcription factors and is active in the lymphoid system (Speck et al, 1990; Fig 2A). The inserted MoMLV enhancer induced active chromatin at the 3′ end of the Prdm1 gene in Prdm1ihCd2/+ pro-B cells (Appendix Fig S2A) and led to a 86- and 58-fold increase of Prdm1 mRNA expression in Prdm1ihCd2/+ pro-B and pre-B cells compared to wild-type counterpart cells, as determined by RNA-seq (Fig 2B). RT–qPCR analysis revealed that the nascent Prdm1 transcripts were increased 26- to 28-fold in Prdm1ihCd2/+ pre-B cells relative to wild-type pre-B cells (Fig 2C). Moreover, the level of nascent Prdm1 transcripts in Prdm1ihCd2/+ pre-B cells was only 3.4- to 4-fold below that of LPS-induced plasmablasts of both the Prdm1ihCd2/+ and wild-type genotype, which furthermore revealed that the MoMLV enhancer did not contribute to the high transcription rate of Prdm1 in plasmablasts (Fig 2C). Notably, the inserted MoMLV enhancer did not affect Atg5 transcription and mRNA expression (Fig 2B and C) and was part of the Prdm1-ihCd2 transcript that still contained the 3′ UTR sequence of the Prdm1 gene (Appendix Fig S2B). Moreover, Flpe-mediated deletion of the frt-flanked sequences containing the MoMLV enhancer and ihCd2 reporter gene (Fig 2A) restored normal physiological expression of Prdm1 in the B cell lineage (Minnich et al, 2016). In summary, these data demonstrate that the inserted MoMLV enhancer strongly activated Prdm1 transcription and mRNA expression in Prdm1ihCd2/+ pro-B and pre-B cells.

We next analyzed the expression of the hCD2 protein, reporting Prdm1 mRNA expression, at different B cell developmental stages in Prdm1ihCd2/+ mice by flow cytometric analysis. hCD2 expression was already detected in uncommitted lymphoid progenitors (LMPPs, ALPs, BLPs); was increased in pro-B, pre-B, and immature B cells of the bone marrow; was reduced in marginal zone (MZ) and follicular (FO) B cells of the spleen as well as in B-1 cells of the peritoneal cavity; and, as expected, was most highly activated in splenic plasma cells (Fig 2D and Appendix Fig S2C). hCD2 expression was low in double-negative (DN) thymocytes, increased in double-positive (DP) thymocytes, and was still observed in splenic CD4 and

CD8 T cells as well as in natural killer (NK) cells, whereas hCD2 expression was absent in granulocytes and macrophages (Fig 2E, Appendix Fig S2D and data not shown). Next, we directly investigated expression of the Blimp1 protein by intracellular staining (Fig 2D and E, and Appendix Fig S2C and D), which was, however, less sensitive compared to the hCD2 analysis, but closely correlated with the hCD2 expression pattern (Appendix Fig S2E). Blimp1 expression was readily detectable in BLPs, pro-B, pre-B, immature B cells, and plasma cells of the bone marrow as well as in DP thymocytes of Prdm1ihCd2/+ mice (Fig 2D and Appendix Fig S2D) and was confirmed for pro-B cells by immunoblotting (Fig 2F). We next generated Prdm1Gfp/ihCd2 mice to examine whether ectopic Blimp1 expression from the Prdm1ihCd2 allele may auto-regulate and thus activate expression of the second Prdm1Gfp allele (Kallies et al, 2004). Despite hCD2 expression, the pro-B, pre-B, and immature B cells of Prdm1Gfp/ihCd2 mice did not express GFP (Appendix Fig S2F), demonstrating that Blimp1-mediated auto-regulation did not occur in these B cell subsets. Collectively, these data demonstrate that the MoMLV enhancer insertion in the Prdm1 locus resulted in premature Blimp1 expression during B cell development and, to a lower degree, in mature T cells.

## Impaired B cell development in Prdm1ihCd2/+ mice

Ectopic expression of Blimp1 resulted in a 2.4-fold decrease in total B cells (CD19+B220+) in the bone marrow of Prdm1ihCd2/+ mice compared to wild-type mice, as shown by flow cytometry (Fig 3A). Surprisingly however, B cells were almost completely lost in homozygous Prdm1ihCd2/ihCd2 mice (Fig 3A), indicating that a further increase of ectopic Blimp1 expression effectively disrupted B cell development. While pro-B cells (KithiCD19+CD25−IgM−IgD−) were present at similar numbers in the bone marrow of Prdm1ihCd2/+ mice, pre-B (Kit−CD19+CD25+IgM−IgD−), immature B (CD19+IgM+IgD−), and recirculating B (CD19+IgMloIgD+) cells were reduced 3.2-, 2.5-, and 8-fold, respectively (Fig 3B). Moreover, splenic MZ (CD19+CD21hiCD23lo) and FO (CD19+CD21intCD23hi) B cells as well as peritoneal B-1a cells (CD5+CD19+CD23−) were decreased 2.6-, 6.9-, and 21-fold, respectively, in Prdm1ihCd2/+ mice compared to wild-type

---

**Figure 2. Premature expression of the Blimp1 protein in Prdm1ihCd2/+ lymphocytes.**

A Schematic diagram of the 3′ end of the Prdm1ihCd2 allele (Minnich et al, 2016). The frt-flanked IRES-hCd2 (ihCD2) reporter gene, which was linked to a 258-bp long terminal repeat (LTR) sequence of the Moloney murine leukemia virus (MoMLV, green), was inserted between the stop codon and 3′ UTR of the Prdm1 gene. C-terminal sequences of exon 8 contained in-frame tag sequences encoding the Flag and V5 epitopes, two TEV protease cleavage sites, and a biotin acceptor sequence. The inserted MoMLV enhancer sequence is shown below together with its regulatory elements (Speck et al, 1990). GRE, glucocorticoid response element; LV, leukemia virus factor-binding site; NF1, nuclear factor 1; polyadenylation site, pA.

B Expression of Prdm1 and Atg5 mRNA in ex vivo sorted pro-B and pre-B cells from the bone marrow of Prdm1ihCd2/+ or wild-type (WT) mice. mRNA expression is shown as mean expression value (TPM) with SEM based on two independent RNA-seq experiments for each cell type and genotype.

C RT–qPCR analysis of nascent Prdm1 and Atg5 transcripts in ex vivo sorted pre-B cells and in in vitro differentiated plasmablasts (PB) of the indicated genotypes. Plasmablasts were generated by stimulation of splenic FO B cells for 4 days with LPS. The data were normalized to those of the ubiquitously expressed control gene Tbp and are presented relative to those of the wild-type pre-B cells (set as 1). Nascent transcripts were PCR-amplified with primers located in the indicated introns (Table EV3). Statistical data are shown as mean value with SEM and were analyzed by the Student's t-test; *P < 0.05, **P < 0.01. Each dot corresponds to one mouse.

D, E Flow cytometric analysis of hCD2 cell surface expression and intracellular Blimp1 expression of the indicated cell types. Bone marrow of Prdm1ihCd2/+ (red) or wild-type (WT, gray) mice was used to analyze ALPs, BLPs, pro-B, pre-B, immature B cells, NK cells, and granulocytes, whereas the spleen was used for flow cytometric analysis of FO B, MZ B, and plasma cells as well as naïve CD4 T and naïve CD8 T cells of both genotypes. The histograms show hCD2 (top row) and Blimp1 (bottom row) expression for the different cell types, which were defined as described in the Appendix Supplementary Methods. Wild-type FO B cells (dashed line) were used as negative control for the Blimp1 staining in plasma cells. The difference in mean fluorescence intensity (ΔMFI) between the two genotypes is shown for each cell type.

F Immunoblot analysis of Blimp1 and the TATA-binding protein (TBP) in nuclear extracts prepared from short-term cultured wild-type or Prdm1ihCd2/+ pro-B cells. Marker proteins of the indicated size (in kilodaltons) are shown to the right.

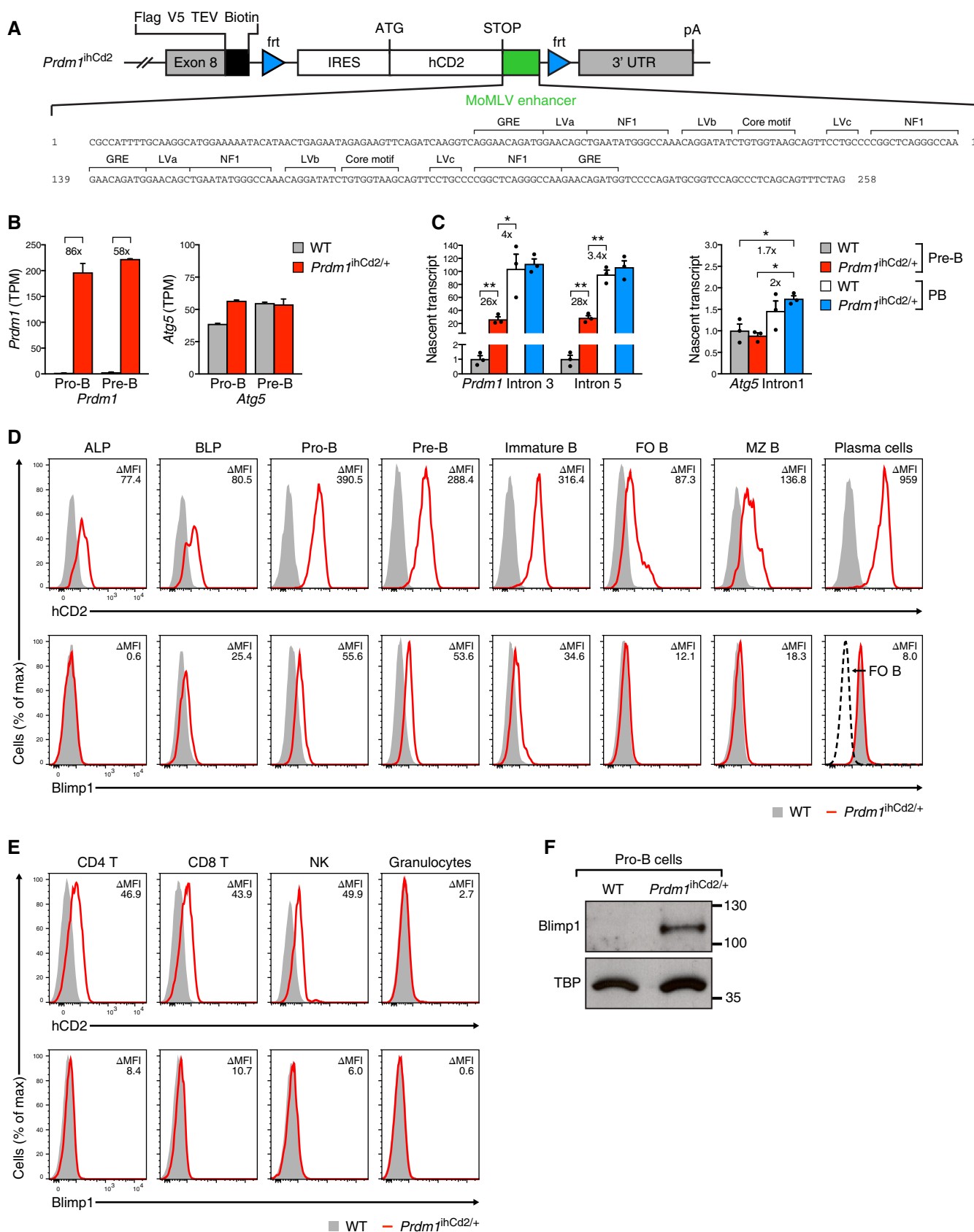

**Figure 2.**

mice (Fig 3C and Appendix Fig S3A). We next investigated whether the observed developmental defects were caused by apoptosis. Notably, the analysis of *ex vivo* *Prdm1*[ihCd2/+] pro-B cells by two different apoptosis assays did not reveal increased cell death compared to wild-type pro-B cells (Fig 3D and Appendix Fig S3B), consistent with the observed similar abundance of *Prdm1*[ihCd2/+] and wild-type pro-B cells in the bone marrow (Fig 3B). In contrast, *Prdm1*[ihCd2/+] pre-B cells exhibited a 2-fold increase in apoptosis relative to wild-type pre-B cells (Fig 3D and Appendix Fig S3B). Interestingly, B cell development in the bone marrow and, to a lower degree, in the spleen was rescued in *Vav-Bcl2 Prdm1*[ihCd2/+] mice (Appendix Fig S3C), which constitutively expressed the pro-survival protein Bcl2 from the *Vav-Bcl2* transgene in all hematopoietic cell types (Ogilvy *et al*, 1999). Hence, apoptosis contributes to the B cell development defects observed in *Prdm1*[ihCd2/+] mice.

To measure the *in vivo* lifespan of Blimp1-expressing FO B cells, we continuously labeled *Prdm1*[ihCd2/+] and wild-type mice with the thymidine analogue bromodeoxyuridine (BrdU) for 10 days prior to flow cytometric analysis of BrdU incorporation in FO B cells (Fig 3E and F). As previously published (Rolink *et al*, 1998), BrdU was incorporated in only 10% of all FO B cells in contrast to 68% of the immature B cells (CD21⁻CD23⁻B220⁺CD19⁺) in the spleen of control wild-type mice (Fig 3E and F), indicating that few immature B cells were recruited into the quiescent FO B cell pool during the 10-day labeling period. In contrast, 21% (2.1-fold increase) of the splenic FO B cells in *Prdm1*[ihCd2/+] mice incorporated BrdU during the first 10 days, but half of them were then replaced by unlabeled *Prdm1*[ihCd2/+] FO B cells during the subsequent 15-day chase period in contrast to the wild-type FO B cells (Fig 3E and F). These data therefore revealed a shortened lifespan and thus more rapid turnover of *Prdm1*[ihCd2/+] FO B cells compared to control FO B cells in the spleen.

The development of T and NK cells was only moderately affected even in homozygous *Prdm1*[ihCd2/ihCd2] mice (Appendix Fig S3D–F). We conclude therefore that ectopic Blimp1 expression from the *Prdm1*[ihCd2] allele preferentially impaired B lymphopoiesis.

## Blimp1-dependent deregulation of the B cell gene expression program

As an important role of Blimp1 in plasma cells is to suppress the B cell gene expression program (Shaffer *et al*, 2002; Minnich *et al*,

2016), we investigated the degree to which the ectopically expressed Blimp1 protein could interfere with the transcriptional program of developing B cells in *Prdm1*[ihCd2/+] mice. By comparing the gene expression changes between *Prdm1*[ihCd2/+] and wild-type pro-B cells, we identified 208 Blimp1-activated and 113 Blimp1-repressed genes, based on an expression difference of > 3-fold, an adjusted *P*-value of < 0.05, and an expression value of > 5 TPM (transcripts per million) in one of the two pro-B cell types (Fig 4A and Table EV1). The same expression analysis identified 280 Blimp1-activated and 125 Blimp1-repressed genes in *Prdm1*[ihCd2/+] pre-B cells (Fig 4B and Table EV1) with an overlap of 67 activated and 42 repressed genes between pro-B and pre-B cells (Fig 4C and Appendix Fig S4A). Notably, the comparison of Blimp1-regulated genes between *Prdm1*[ihCd2/+] pro-B or pre-B cells and wild-type pre-plasmablasts (Minnich *et al*, 2016) revealed an overlap of 12 or 38 Blimp1-activated and 17 or 36 Blimp1-repressed genes (Appendix Fig S4A), respectively, indicating that some genes were similarly regulated by Blimp1 in early B cells and plasmablasts.

As the ectopically expressed Blimp1 protein contained a V5 epitope sequence (Fig 2A), we determined the genome-wide Blimp1-binding pattern in *Prdm1*[ihCd2/+] pro-B cells by ChIP-seq analysis with an anti-V5 antibody. Peak calling with a stringent *P*-value of < 10⁻¹⁰ identified 762 Blimp1 peaks with a consensus Blimp1-binding motif in *Prdm1*[ihCd2/+] pro-B cells (Fig 4D and Appendix Fig S4B). Although the Blimp1 peaks in *Prdm1*[ihCd2/+] pro-B cells were 12-fold reduced in number compared to the Blimp1-bound sites (9,320) in plasmablasts (Minnich *et al*, 2016), the majority (88%) of them were also present in plasmablasts, but exhibited a 2.5-fold lower Blimp1-binding density compared to the corresponding Blimp1 peaks in plasmablasts (Fig 4D and E). Blimp1 binding was observed at one-third of all repressed genes in *Prdm1*[ihCd2/+] pro-B or pre-B cells (Fig 4F and G), which resulted in nine commonly repressed Blimp1 target genes in Blimp1-expressing pro-B cells, pre-B cells, and plasmablasts (Fig 4G and Appendix Fig S4C and D). In contrast, Blimp1 binding was detected at only 4.3% of all activated genes in *Prdm1*[ihCd2/+] pro-B or pre-B cells (Fig 4F), which resulted in the identification of only one commonly activated target gene in Blimp1-expressing pre-B cells and plasmablasts (Appendix Fig S4C). We conclude therefore that Blimp1 regulated gene expression in pro-B and pre-B cells primarily in an indirect manner, which is in marked contrast to the high proportion of

---

**Figure 3. Impaired B cell development in *Prdm1*[ihCd2/+] mice.**

A, B  Flow cytometric analysis of total B cells (A) as well as pro-B, pre-B, immature (imm) B, and recirculating (recirc) B cells (B) from the bone marrow of wild-type (WT, gray), *Prdm1*[ihCd2/+] (red), and *Prdm1*[ihCd2/ihCd2] (white) mice at the age of 6-8 weeks.

C     Flow cytometric analysis of total B, FO B, and MZ B cells from the spleens of wild-type, *Prdm1*[ihCd2/+], and *Prdm1*[ihCd2/ihCd2] mice. Bar graphs show absolute cell numbers for each cell type and indicated genotype. The different cell types were defined as described in detail in the Appendix Supplementary Methods. The total numbers of bone marrow cells and splenocytes are shown in Appendix Fig S5A.

D     Determination of cell death of *ex vivo* pro-B and pre-B cells from the bone marrow of *Prdm1*[ihCd2/+] (red) and wild-type (gray) mice by flow cytometric analysis of the loss of plasma membrane asymmetry (F2N12S ratiometric dye staining) and the loss of cell membrane integrity (SYTOX staining). Representative flow cytometry plots (left) and the quantification of dead and apoptotic (apop) cells (right) are shown.

E, F  BrdU labeling of immature (imm) and FO B cells in the spleen of 3-month-old *Prdm1*[ihCd2/+] mice (red dots) and control wild-type littermates (gray dots). The percentage of BrdU⁺ B cells was determined for each B cell type by flow cytometric analysis after 10 days of continuous BrdU labeling (day 10, white bar) or after a subsequent 15-day chase period (day 25; hatched bar) without BrdU in the drinking water (E). A diagram (below) indicates the design of the BrdU labeling and chase experiments. Flow cytometric data obtained with FO B cells from one mouse of each genotype are shown in (F). The percentage of BrdU⁺ FO B cells is shown above the indicated gates (F).

Data information: Statistical data (A–E) are shown as mean value with SEM and were analyzed by the Student's *t*-test; *P* < 0.05, ***P* < 0.001, ****P* < 0.0001. Each dot corresponds to one mouse.

  

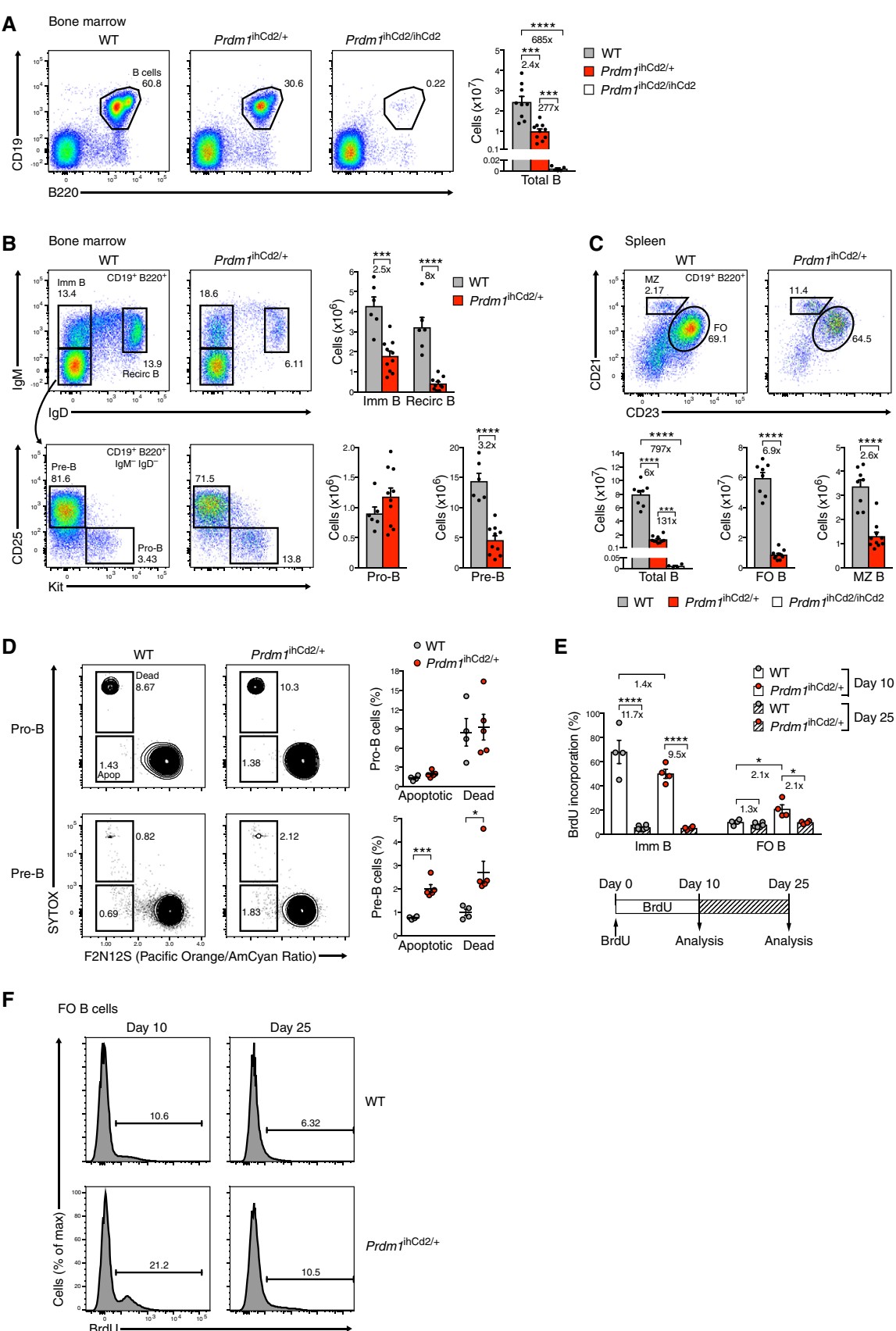

**Figure 3.**

directly regulated Blimp1 target genes identified in plasmablasts (Minnich *et al*, 2016).

Annotation of the Blimp1-regulated genes in *Prdm1*[ihCd2/+] pre-B cells revealed that half of these genes coded for proteins of the following functional classes: 45 activated and 23 repressed cell surface proteins, 51 activated and 18 repressed signal transducers, 24 activated and 10 repressed transcriptional regulators, and 21 activated and 13 repressed metabolic enzymes (Appendix Fig S4E and Table EV1). The deregulated transcriptional regulators likely explain the indirect control of gene expression by Blimp1 in pro-B and pre-B cells (Fig 4H and Table EV1). Notably, the B cell commitment gene *Pax5* was 2.8-fold repressed in *Prdm1*[ihCd2/+] pre-B cells (Fig 4H), which was observed only at the pre-B and immature B cell stages of B lymphopoiesis (Appendix Fig S4F). The deregulation of multiple cell surface receptors and intracellular signal transducers suggested that Blimp1 affected the normal signaling responses of B cells (Fig 4I and Appendix Fig S4G). In summary, these data indicate that premature Blimp1 expression in *Prdm1*[ihCd2/+] mice strongly interfered with the B cell gene expression program controlling B cell development.

## Increased plasma cell development in the absence of GC B cells in *Prdm1*[ihCd2/+] mice

As ectopic Blimp1 expression interfered with the B-lymphoid gene expression program in B cells in a manner similar to Blimp1's physiological role in plasma cells (Minnich *et al*, 2016), we next investigated whether premature Blimp1 expression could lead to increased plasma cell differentiation in *Prdm1*[ihCd2/+] mice. Indeed, plasma cells (Lin⁻B220[int/−]CD138⁺CD28⁺) were significantly increased in the spleen and bone marrow of non-immunized *Prdm1*[ihCd2/+] mice compared to wild-type mice at the age of 2–12 months (Fig 5A and B, and Appendix Fig S5A). Accordingly, the serum titers of IgM and IgG antibodies (measured by ELISA) as well as the number of plasma cells secreting different IgG isotype antibodies (determined by ELISPOT assay) were also increased in *Prdm1*[ihCd2/+] mice compared to wild-type littermates (Appendix Fig S5B and C). Interestingly, FO B cells in the spleen of non-immunized *Prdm1*[ihCd2/+]

mice already expressed higher levels of the activation markers CD40, CD80, CD86, and MHCII (Appendix Fig S5D), suggesting that the altered threshold for B cell activation may contribute to enhanced plasma cell differentiation in *Prdm1*[ihCd2/+] mice. We next immunized experimental and control mice with 4-hydroxy-3-nitro-phenylacetyl-conjugated keyhole limpet hemocyanin (NP-KLH). At day 14 after immunization, anti-NP-IgM antibody-secreting cells (ASCs) were 4.6-fold increased in the spleen of *Prdm1*[ihCd2/+] mice, as determined by ELISPOT assay (Fig 5C), and the titers of anti-NP-IgM were 2.4-fold higher in the serum of *Prdm1*[ihCd2/+] mice compared to wild-type littermates, as measured by ELISA (Fig 5D). Notably, the increase of plasma cells was observed despite a 6-fold decrease of B cells in the presence of a similar number of total splenocytes in *Prdm1*[ihCd2/+] mice relative to wild-type littermates (Fig 3C and Appendix Fig S5E). We conclude therefore that plasma cell development was strongly enhanced also in immunized *Prdm1*[ihCd2/+] mice.

In contrast to the increased anti-NP-IgM levels, the serum titers of anti-NP-IgG1 and anti-NP-IgG2b were decreased in immunized *Prdm1*[ihCd2/+] mice compared to control littermates (Fig 5D). Moreover, the frequency of somatic hypermutation (SHM) at the rearranged *Igh* gene was reduced in splenic plasma cells of *Prdm1*[ihCd2/+] mice compared to wild-type littermates (Appendix Fig S5F). As suggested by the observed reduction of SHM, germinal center (GC) B cells (CD19⁺B220⁺GL7⁺Fas⁺) were strongly decreased in the spleen of *Prdm1*[ihCd2/+] mice relative to control mice at day 14 after NP-KLH immunization (Fig 5E) as well as in non-immunized mice (Appendix Fig S5G). Moreover, the few residual *Prdm1*[ihCd2/+] GC B cells did not express hCD2 (Blimp1) and revealed normal expression of the transcription factor Bcl6 (Fig 5F), which is an essential regulator of GC B cell differentiation (Dent *et al*, 1997; Ye *et al*, 1997). Notably, follicular helper T (T_FH) cells (CXCR5⁺PD-1⁺CD4⁺B220⁻), which provide T cell help to support GC B cell differentiation (Vinuesa *et al*, 2016), were 3.3-fold decreased in the spleen of immunized *Prdm1*[ihCd2/+] mice relative to wild-type mice and expressed hCD2 (Blimp1), albeit at different levels (Fig 5G and H). The hCD2[lo] population of the *Prdm1*[ihCd2/+] T_FH cells expressed little Blimp1 protein and minimally reduced levels of Bcl6 (Fig 5H),

**Figure 4.  Blimp1-dependent deregulation of the B cell gene expression program.**

A, B   Scatter plot of gene expression differences in *ex vivo* sorted *Prdm1*[ihCd2/+] and wild-type (WT) pro-B (A) and pre-B (B) cells, based on two RNA-seq experiments for each cell type and genotype. The expression data of individual genes (indicated by dots) were plotted as normalized (norm) rlog values. Genes with an expression difference of > 3-fold, an adjusted *P*-value of < 0.05, and a TPM value of > 5 (in one of the two pro-B or pre-B cell types, respectively) are colored in blue or red, corresponding to activation or repression by Blimp1.

C   Expression of commonly activated (blue) and commonly repressed (red) genes in pro-B and pre-B cells. The log₂-fold expression change observed between *Prdm1*[ihCd2/+] and wild-type pro-B cells (horizontal axis) as well as between *Prdm1*[ihCd2/+] and wild-type pre-B cells (vertical axis) is plotted for each gene. Lines indicated 2-fold (2×) and 3-fold (3×) expression differences.

D   Identification of 762 Blimp1 peaks in short-term cultured *Prdm1*[ihCd2/+] pro-B cells, as detected by ChIP-seq with an anti-V5 antibody and MACS peak calling with a *P*-value of < 10⁻¹⁰. The 9,320 Blimp1 peaks identified in plasmablasts (PB; Minnich *et al*, 2016) are shown for comparison. Common (black) and unique (white) peaks are shown for both cell types.

E   Densities of Blimp1 binding. Average read density profiles aligned at the center of the Blimp1 peak are shown for the common Blimp peaks of both cell types.

F   Blimp1 binding at activated and repressed genes in *Prdm1*[ihCd2/+] pro-B and pre-B cells. Regulated genes, which are bound by Blimp1 in *Prdm1*[ihCd2/+] pro-B cells, are shown in black.

G   Blimp1 binding and regulation of the commonly repressed target genes *Sell* (CD62L) and *B3gnt5*. The ChIP-seq data (left) were obtained with *Prdm1*[ihCd2/+] pro-B cells (this study) and *Prdm1*[Bio/Bio] *Rosa26*[BirA/BirA] plasmablasts (PB; Minnich *et al*, 2016). The RNA-seq data (right) were determined in pro-B and pre-B cells of the *Prdm1*[ihCd2/+] (red) and wild-type (gray) genotype and in Blimp1-deficient (*Prdm1*[Gfp/Δ], blue) and wild-type (gray) pre-plasmablasts (Pre-PB; Minnich *et al*, 2016). Cell types expressing Blimp1 (+) are indicated.

H, I   Expression of selected Blimp1-activated and Blimp1-repressed genes coding for transcriptional regulators (H) and intracellular signal transducers (I) in *Prdm1*[ihCd2/+] (red) and wild-type (gray) pre-B cells. Blimp1-bound genes are underlined. The mRNA expression of the indicated genes is shown as mean expression value (TPM) with SEM, based on two different RNA-seq experiments for pre-B cells of each genotype.

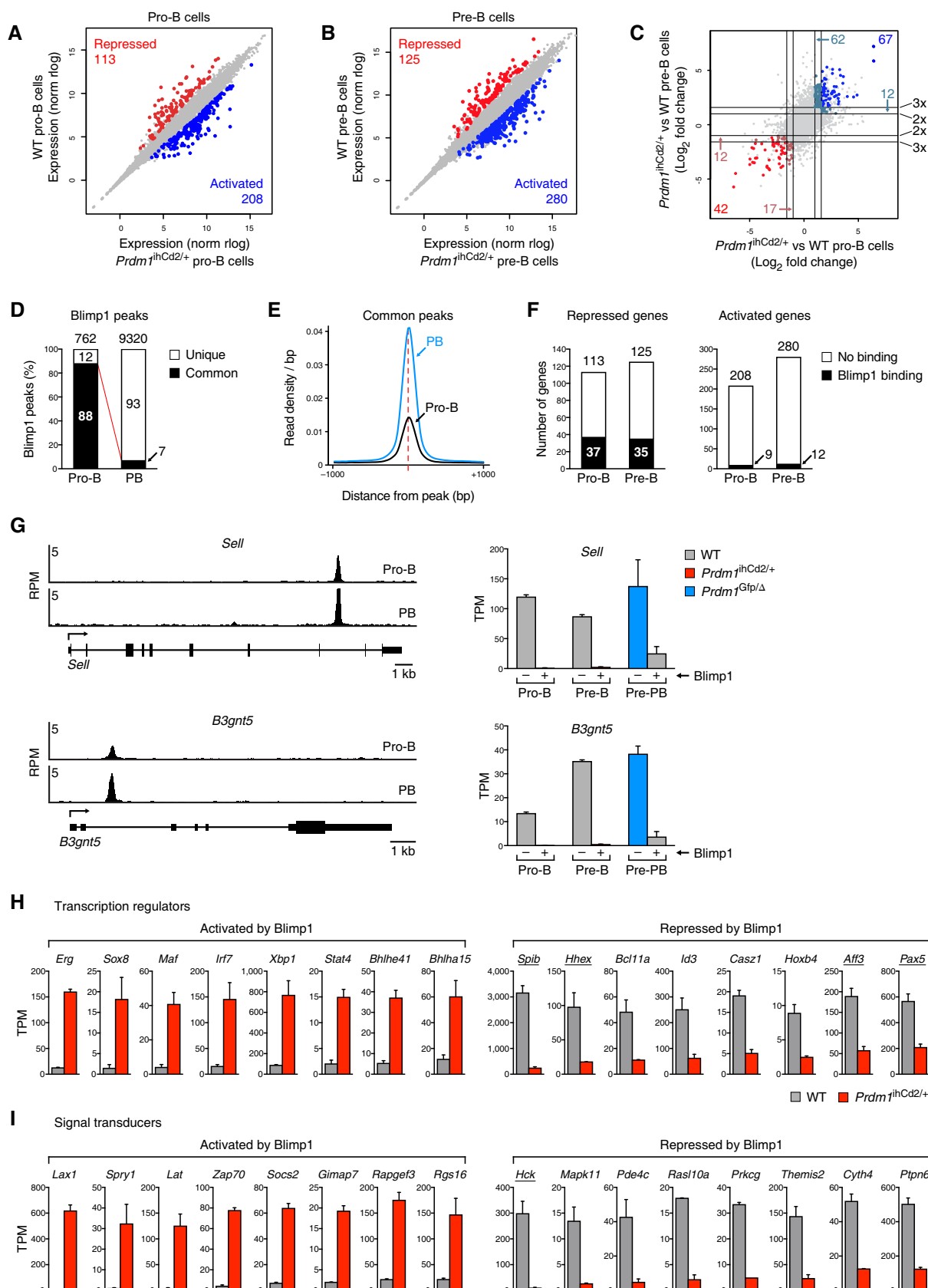

**Figure 4.**

which is also an essential regulator of $T_{FH}$ cell differentiation (Johnston *et al*, 2009; Nurieva *et al*, 2009; Yu *et al*, 2009). In contrast, the hCD2$^{hi}$ subset of *Prdm1*$^{ihCd2/+}$ $T_{FH}$ cells expressed Blimp1, leading to strongly decreased Bcl6 expression (Fig 5H), which suggests that the ectopically expressed Blimp1 protein repressed *Bcl6* in this subset, consistent with an antagonistic role of these transcription factors in $T_{FH}$ cells (Johnston *et al*, 2009). Hence, these data imply that ectopic Blimp1 expression suppresses $T_{FH}$ cell differentiation by interfering with the Bcl6-regulated gene expression program, which likely contributes to the loss of GC B cells in *Prdm1*$^{ihCd2/+}$ mice (Johnston *et al*, 2009; Nurieva *et al*, 2009; Yu *et al*, 2009).

### Blimp1 strongly enhances plasmablast differentiation of immature and mature B cells

We next directly investigated whether precocious Blimp1 expression may endow immature and mature B cells with an enhanced potential to differentiate to plasmablasts. To this end, we stimulated immature B cells (CD19$^+$B220$^+$IgM$^+$IgD$^-$) from the bone marrow of *Prdm1*$^{ihCd2/+}$ or control wild-type mice with CpG oligodeoxynucleotides for 60 h, as activation of the Toll-like receptor 9 (TLR9) is known to promote differentiation of immature B cells to IgM-secreting plasmablasts (Azulay-Debby *et al*, 2007). Interestingly, the immature B cells of *Prdm1*$^{ihCd2/+}$ mice underwent differentiation to plasmablasts (CD138$^+$CD22$^{lo}$) at a 8.3-fold higher frequency than wild-type immature B cells (Fig 5I). Likewise, lipopolysaccharide (LPS)-mediated TLR4 activation of *Prdm1*$^{ihCd2/+}$ immature B cells resulted in a 3.8-fold increase of plasmablast formation compared to that of wild-type immature B cells (Fig 5I). These findings were confirmed by the presence of Igκ-containing antibodies at 8.7- and 18-fold higher levels in the supernatant of the CpG- and LPS-stimulated *Prdm1*$^{ihCd2/+}$ plasmablasts compared to identically treated wild-type plasmablasts (Fig 5J). Finally, mature FO B cells from the spleen of *Prdm1*$^{ihCd2/+}$ mice also differentiated more efficiently to plasmablasts than wild-type FO B cells after 4 days of stimulation with

CpG, LPS or a combination of IL-4, IL-5, and anti-CD40 (Appendix Fig S5H–J). We conclude therefore that precocious Blimp1 expression strongly enhanced the plasmablast differentiation potential of the first IgM$^+$ immature B cells in the bone marrow as well as of mature FO B cells in the spleen.

### Blimp1 expression in $T_{FH}$ cells causes the loss of GC B cells in *Prdm1*$^{ihCd2/+}$ mice

As CD8 and CD4 T cells including $T_{FH}$ cells ectopically expressed Blimp1 in *Prdm1*$^{ihCd2/+}$ mice (Figs 2E and 5H), we next examined whether the B cell phenotype of these mice is B cell-intrinsic or depends on Blimp1-expressing T cells. To this end, we generated mixed bone marrow chimeras by reconstituting lethally irradiated *Rag2*$^{-/-}$ mice with a mixture of Eβ$^{-/-}$ *Prdm1*$^{ihCd2/+}$ and J$_H$T *Prdm1*$^{+/+}$ bone marrow at a ratio of 15:1 (Fig 6A). A higher amount of the *Prdm1*$^{ihCd2/+}$ bone marrow was required as ectopic Blimp1 expression in uncommitted lymphoid progenitors (LMPPs, ALPs, BLPs; Fig 2D and Appendix Fig S2C) conferred a competitive disadvantage to the *Prdm1*$^{ihCd2/+}$ progenitors relative to the J$_H$T *Prdm1*$^{+/+}$ progenitors (data not shown). All B cells in the chimeric mice originated from the Eβ$^{-/-}$ *Prdm1*$^{ihCd2/+}$ stem cells, as the homozygous J$_H$T mutation, eliminating the D$_H$Q52, J$_H$, and Eμ elements of the *Igh* locus (Gu *et al*, 1993), prevented B cell development of the J$_H$T *Prdm1*$^{+/+}$ progenitors. In contrast, all T cells were derived from J$_H$T *Prdm1*$^{+/+}$ stem cells, as the Eβ$^{-/-}$ *Prdm1*$^{ihCd2/+}$ progenitors could not contribute to αβ T cell development due to deletion of the Eβ enhancer of the *Tcrb* locus (Bouvier *et al*, 1996). For comparison, we generated bone marrow chimeras with a mixture of Eβ$^{-/-}$ *Prdm1*$^{+/+}$ and J$_H$T *Prdm1*$^{+/+}$ bone marrow and analyzed all chimeric mice 4 months after transplantation (Fig 6A). As the lower competitive fitness of the Eβ$^{-/-}$ *Prdm1*$^{ihCd2/+}$ progenitors compared to the control Eβ$^{-/-}$ *Prdm1*$^{+/+}$ progenitors could also contribute to the observed difference in B cell development, we analyzed the trend, but not the absolute fold

---

**Figure 5.** Increased plasma cell development and decreased $T_{FH}$ cell differentiation in the absence of GC B cell formation in *Prdm1*$^{ihCd2/+}$ mice.

A, B    Flow cytometric analysis of plasma cells from the spleen (A) and bone marrow (B) of *Prdm1*$^{ihCd2/+}$ and wild-type (WT) mice under steady-state conditions. Representative flow cytometric data are shown for 12-month-old mice (left), while bar graphs present absolute numbers of plasma cells for *Prdm1*$^{ihCd2/+}$ (red) and wild-type (gray) mice at the age of 2, 4, and 12 months (right). The total number of cells in the spleen and bone marrow of the analyzed mice is shown in Appendix Fig S5A.

C, D    T cell-dependent immune responses. *Prdm1*$^{ihCd2/+}$ (red dots) and wild-type (gray dots) mice at the age of 2 months were immunized with NP-KLH (in alum) and analyzed at day 14 after immunization by ELISPOT assay (C) and ELISA (D). The number of anti-NP-IgM antibody-secreting cells (ASCs) in the spleen was determined by ELISPOT assay (C) using NP$_{24}$-BSA-coated plates for detecting cells secreting total anti-NP-IgM antibodies. ELISPOT results are shown as representative pictures (left) or absolute ASC numbers (right) (C). The serum titers of anti-NP-specific IgM, IgG1, and IgG2b antibodies were analyzed by ELISA using NP$_7$-BSA- or NP$_{24}$-BSA-coated plates for detecting high-affinity IgG1 or total IgM, IgG1, and IgG2b antibodies, respectively (D). Dot plots display the serum immunoglobulin titers of *Prdm1*$^{ihCd2/+}$ (red dots) and wild-type (gray dots) mice. NP-specific IgG1 concentrations (μg/ml) were determined relative to a standard NP-binding IgG1 antibody, whereas IgM and IgG2b amounts are indicated as relative units (RU) by setting the WT data to 1. The total number of splenocytes in the immunized mice is shown in Appendix Fig S5E.

E, F    Flow cytometric analysis of splenic GC B cells in *Prdm1*$^{ihCd2/+}$ (red) and wild-type (gray) mice (at the age of 2 months) at day 14 after NP-KLH immunization. Bar graphs display absolute numbers of GC B cells detected in the spleen of both genotypes (E). Cell surface expression of hCD2 and intracellular staining of Bcl6 are shown for *Prdm1*$^{ihCd2/+}$ (red) and wild-type (gray) GC B cells (F). The absence of Bcl6 expression in non-GC B cells (black) is shown as reference.

G, H    Flow cytometric analysis of splenic $T_{FH}$ cells from the same mice analyzed in (E, F). Bar graphs indicate absolute numbers of $T_{FH}$ cells (G), and histograms display cell surface expression of hCD2 and intracellular staining of Bcl6 (H) in hCD2$^{lo}$ (black) and hCD2$^{hi}$ (blue) *Prdm1*$^{ihCd2/+}$ as well as wild-type (gray) $T_{FH}$ cells.

I    Efficient *in vitro* plasmablast differentiation of immature B cells. Immature B cells sorted from the bone marrow of *Prdm1*$^{ihCd2/+}$ (red) and wild-type (gray) mice were stimulated with CpG oligodeoxynucleotides or LPS for 60 h followed by flow cytometric analysis of CD138$^+$CD22$^{lo}$ plasmablasts (PB). Bar graphs (right) summarize the plasmablast data of the CpG and LPS stimulation experiments.

J    ELISA detection of Igκ-containing antibodies in the supernatants of immature B cells at 60 h of CpG or LPS stimulation.

Data information: Statistical data (A–E, G, I, J) are shown as mean value with SEM and were analyzed by the Student's *t*-test; *$P < 0.05$, **$P < 0.01$, ***$P < 0.001$, ****$P < 0.0001$. Each dot corresponds to one mouse.

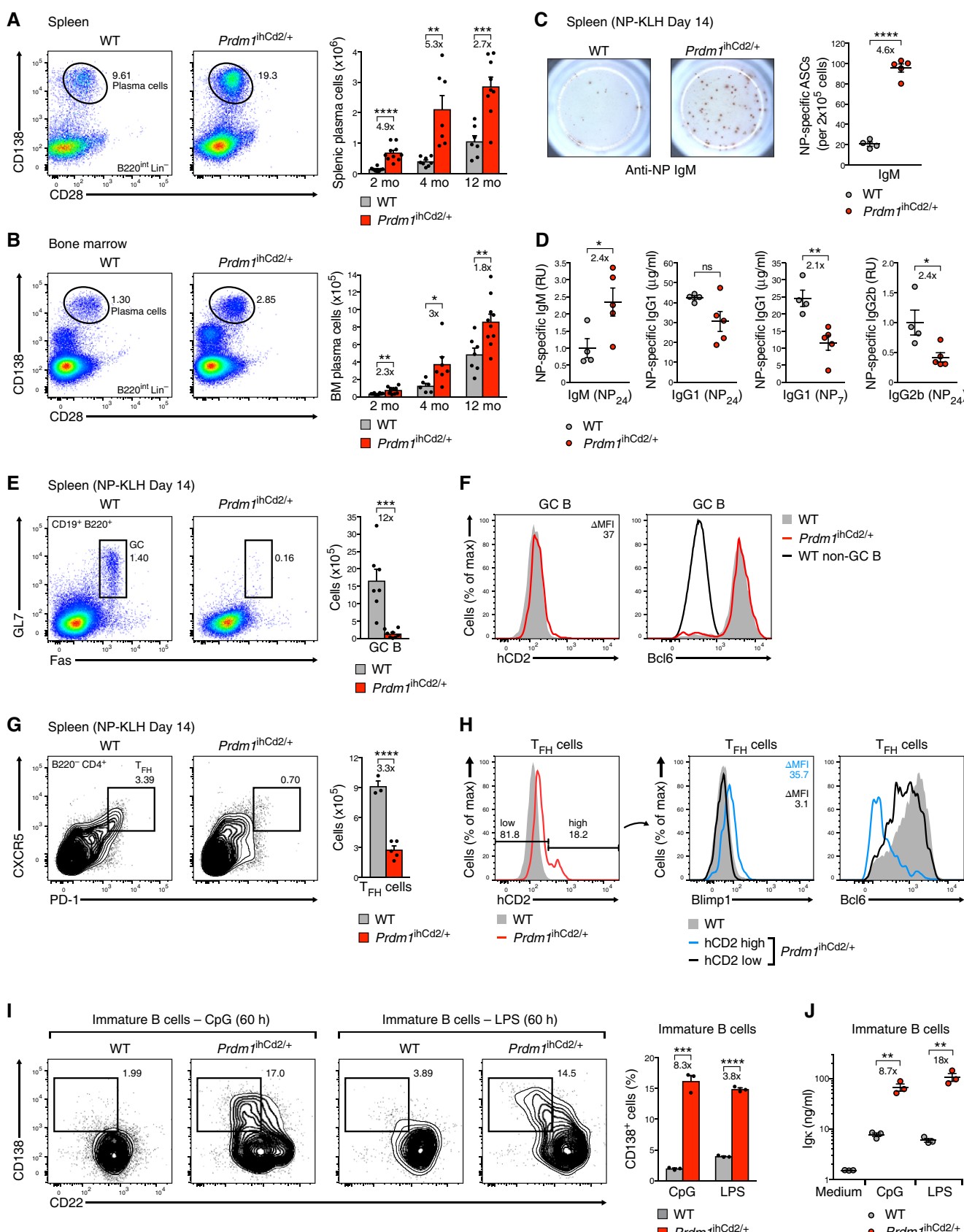

**Figure 5.**

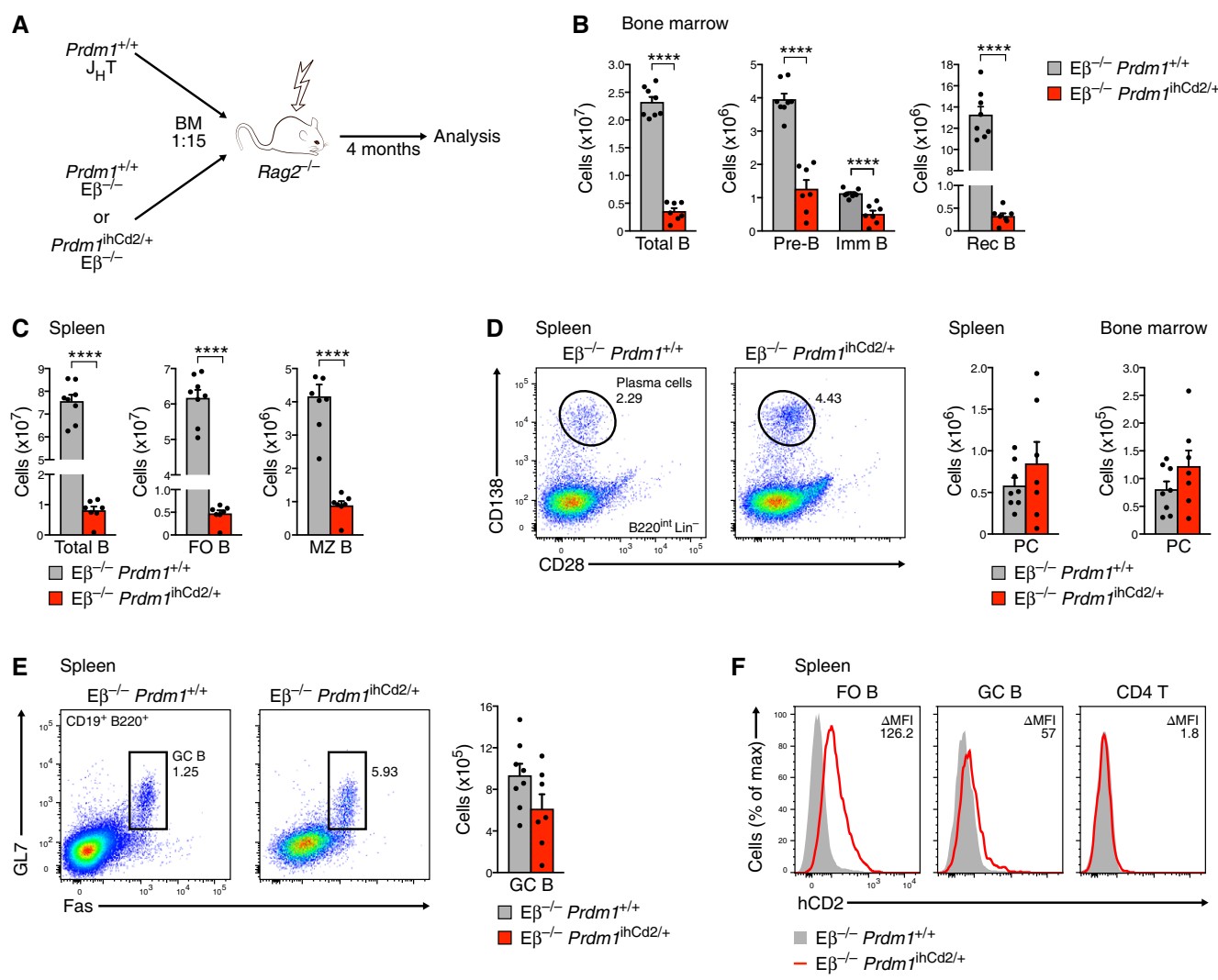

**Figure 6. Cell-autonomous B cell phenotype except for the T cell-dependent loss of GC B cells in *Prdm1*[ihCd2/+] mice.**

A    Schematic diagram describing the generation of mixed bone marrow chimeras. Lineage-depleted bone marrow of B cell-deficient $J_H$T *Prdm1*[+/+] mice was mixed at ratio of 1:15 with lineage-depleted bone marrow of T cell-deficient Eβ[−/−] *Prdm1*[ihCd2/+] or control Eβ[−/−] *Prdm1*[+/+] mice prior to injection into lethally irradiated *Rag2*[−/−] recipient mice and subsequent analysis 4 months after transplantation.

B–F    Flow cytometric analysis of total B cells and different B cell subsets in the bone marrow (B, D) and spleen (C–F) of non-immunized Eβ[−/−] *Prdm1*[ihCd2/+] (red) and Eβ[−/−] *Prdm1*[+/+] (gray) chimeric mice. Bar graphs indicate absolute numbers of the different B cell types in these chimeric mice. (F) Histograms display the expression of hCD2 (Blimp1) in splenic FO B, GC B, and naïve CD4 T cells from the chimeric mice of the indicated genotypes. Statistical data (B–E) are shown as mean value with SEM and were analyzed by the Student's *t*-test; ****$P < 0.0001$. Each dot corresponds to one mouse.

difference of B cell numbers in these chimeras. Whereas all B cell subsets were reduced in the bone marrow and spleen of Eβ[−/−] *Prdm1*[ihCd2/+] chimeric mice compared to the control Eβ[−/−] *Prdm1*[+/+] chimeric mice (Fig 6B and C), plasma cells were slightly increased in both lymphoid organs of the Eβ[−/−] *Prdm1*[ihCd2/+] chimeras (Fig 6D). A similar B cell phenotype was observed by comparing non-chimeric Eβ[−/−] *Prdm1*[ihCd2/+] mice with control Eβ[−/−] *Prdm1*[+/+] mice (Appendix Fig S6A–D). Together, these data indicate that decreased B cell development and increased plasma cell differentiation are a B cell-intrinsic property of the *Prdm1*[ihCd2/+] mouse. However, GC B cells were the exception, as they were present at similar numbers in the spleen of Eβ[−/−] *Prdm1*[ihCd2/+] and control Eβ[−/−] *Prdm1*[+/+] chimeric mice

(Fig 6E) in marked contrast to the loss of GC B cells in *Prdm1*[ihCd2/+] mice (Fig 5E). Since the splenic CD4 T cells in Eβ[−/−] *Prdm1*[ihCd2/+] chimeric mice did not express hCD2 as expected (Fig 6F), we conclude that the ectopic expression of Blimp1 in T cells interfered with the development of GC B cells possibly due to the observed decrease of functional $T_{FH}$ cells in *Prdm1*[ihCd2/+] mice.

## Precocious Blimp1 expression generates an autoimmune disease with progressing age

Given the identification of *PRDM1* as a susceptibility gene for human SLE and RA (Gateva *et al*, 2009; Raychaudhuri *et al*, 2009; Zhou *et al*, 2011), we next investigated whether *Prdm1*[ihCd2/+] mice

develop an autoimmune phenotype with progressing age. Autoimmune diseases, such as SLE, are characterized by circulating autoantibodies that recognize double-stranded (ds) DNA, nuclear proteins [such as SSA (Ro-52) and SSB (La)], and mitochondrial cardiolipin (Suurmond & Diamond, 2015). Hence, we compared, by ELISA, the titers of IgM and IgG recognizing dsDNA, cardiolipin, SSA (Ro-52), and SSB (La) in the serum of *Prdm1*^ihCd2/+ mice with those of the *Fasl*^gld/gld mouse, a known model of severe systemic autoimmune disease (Takahashi *et al*, 1994). The levels of IgM antibodies recognizing dsDNA, cardiolipin, SSA (Ro-52), and SSB (La) were already increased at 2 and 4 months in *Prdm1*^ihCd2/+ mice compared to wild-type mice (Appendix Fig S7A), and elevated titers of the corresponding IgG antibodies were observed at 4 and 12 months in *Prdm1*^ihCd2/+ mice, although they did not reach the high level of the respective IgG antibodies measured in the serum of *Fasl*^gld/gld mice (Fig 7A). Moreover, anti-nuclear antibodies (ANA) of the IgG isotype could readily be detected in the serum of eight out of 22 *Prdm1*^ihCd2/+ mice at the age of 12 months, whereas these autoantibodies were rarely found in the serum of wild-type mice at this age (Fig 7B). Finally, histological analysis of the kidney revealed a significant increase of pathologically altered glomeruli in 12-month-old *Prdm1*^ihCd2/+ mice relative to wild-type mice (Fig 7C and Table EV2). Consistent with these pathological changes, the deposition of IgG immune complexes was detected in glomeruli of the kidneys of some *Prdm1*^ihCd2/+ females (Appendix Fig S7B), further demonstrating that *Prdm1*^ihCd2/+ mice could develop a moderate form of glomerulonephritis.

As defective clearance of apoptotic debris has been causally associated with SLE development (Tsokos *et al*, 2016), we next sought to accelerate the development of the autoimmune phenotype in *Prdm1*^ihCd2/+ mice by five sequential intravenous injections of syngeneic apoptotic thymocytes at weekly intervals (Appendix Fig S8A), as described (Duhlin *et al*, 2016). Ten days after the last injection, *Prdm1*^ihCd2/+ and wild-type mice were analyzed together with non-injected littermates. The serum titers of anti-dsDNA, anti-cardiolipin, anti-SSA (Ro-52), and anti-SSB (La) IgM antibodies were already higher in non-injected *Prdm1*^ihCd2/+ mice compared to wild-type mice (Appendix Fig S8B), as expected for 2-month-old mice (Appendix Fig S7A), but these titers were even further increased in

the injected *Prdm1*^ihCd2/+ mice (Appendix Fig S8B). By contrast, the anti-dsDNA and anti-cardiolipin IgG antibodies were only increased in the injected *Prdm1*^ihCd2/+ mice (Appendix Fig S8B). Finally, while anti-nuclear antibodies of the IgM or IgG isotype were absent in the non-injected *Prdm1*^ihCd2/+ or injected wild-type mice, these antibodies could be readily detected in the serum of three out of four injected *Prdm1*^ihCd2/+ mice (Appendix Fig S8C and data not shown). Together, these data demonstrate that precocious expression of Blimp1 in the B cell lineage predisposes *Prdm1*^ihCd2/+ mice to develop an autoimmune disease with progressing age.

## Discussion

The human *PRDM1* (Blimp1) locus has been identified as a susceptibility gene for development of the autoimmune diseases systemic lupus erythematosus (SLE) and rheumatoid arthritis (RA; Gateva *et al*, 2009; Raychaudhuri *et al*, 2009; Zhou *et al*, 2011). In the B cell lineage, Blimp1 normally acts as an essential regulator of plasma cell development and function (Shapiro-Shelef *et al*, 2003; Kallies *et al*, 2007; Tellier *et al*, 2016). Here, we discovered that the mouse *Prdm1* gene was partially activated at the chromatin and transcription level already in the earliest committed pro-B cells, although mature *Prdm1* mRNA did not accumulate during B cell development due to posttranscriptional regulation. By analyzing a mouse model that facilitated ectopic Blimp1 protein expression throughout B lymphopoiesis, we could demonstrate that Blimp1 impaired B cell development by interfering with the B cell gene expression program, while leading to increased plasma cell differentiation. With progressive aging, these mice developed an autoimmune disease characterized by the appearance of autoantibodies and a moderate form of glomerulonephritis. These data therefore suggest that mutations in the *PRDM1* locus that lead to premature expression of the Blimp1 protein in developing B cells may cause autoimmune diseases such as SLE and RA.

While the expression of key lineage-specific regulators, such as the B cell commitment factor Pax5, is controlled at the transcriptional level (Decker *et al*, 2009), we made the surprising observation that the *Prdm1* locus was already accessible at the chromatin

---

**Figure 7.  Precocious Blimp1 expression causes autoimmunity in *Prdm1*^ihCd2/+ mice with progressing age.**

A   Presence of IgG antibodies detecting dsDNA, cardiolipin, SSA (Ro-52), and SSB (La) in the serum of *Prdm1*^ihCd2/+ (red dots) and wild-type (gray dots) mice at the age of 2, 4, and 12 months. The titers of the different IgG antibodies were determined in the serum of male and female mice by ELISA and are displayed as arbitrary units (AU). The serum of 6-month-old *Fasl*^gld/gld mice was used as positive control.

B   Representative images of anti-nuclear antibody (ANA) staining obtained with serum from 12-month-old *Prdm1*^ihCd2/+ or wild-type mice (right), as detected by indirect immunofluorescence assay using HEp-2 cells and an Alexa488-conjugated anti-mouse IgG detection antibody. The serum of *Fasl*^gld/gld mice was used as positive control. The presence of ANA-IgG antibodies was analyzed for 22 *Prdm1*^ihCd2/+ and 12 wild-type mice, as summarized in the pie chart (right).

C   Periodic acid-Schiff (PAS) staining of paraffin-embedded kidney sections of *Prdm1*^ihCd2/+ and wild-type mice at 12 months of age. The mean score of kidney pathology was determined for each mouse by evaluating 40 individual glomeruli to assess the severity of glomerulonephritis as shown in Table EV2 and described in the Appendix Supplementary Methods. Representative pictures of glomeruli of *Prdm1*^ihCd2/+ kidneys with different pathology scores are shown to the left, and dot plots with average scores for male and female mice (Table EV2) are shown to the right. Arrows denote intracapillary hyaline "thrombi," asterisks indicate mesangial sclerosis with obscured capillary loops, and the arrowhead points to an obsolescent glomerulus with collapse of the glomerular tuft architecture. Statistical data (A, C) are shown as mean value with SEM and were analyzed by the Mann–Whitney $U$-test (A) or the Student's $t$-test (C); *$P < 0.05$, **$P < 0.01$, ***$P < 0.001$. Each dot corresponds to one mouse.

D   Proposed model explaining the autoimmune phenotype of *Prdm1*^ihCd2/+ mice. Precocious Blimp1 expression in *Prdm1*^ihCd2/+ mice results in decreased numbers of developing B cells (red arrows) and increased differentiation of immature and mature B cells to plasmablasts (blue arrows) and plasma cells (green arrow). As a consequence, autoreactive immature B cells may differentiate to plasmablasts prior to their elimination by central tolerance mechanisms (receptor editing or clonal deletion; Nemazee, 2017), and autoreactive mature B cells may escape their anergic state, imposed by anergy-induced BCR desensitization (peripheral tolerance; Theofilopoulos *et al*, 2017), by premature differentiation to plasmablasts, which leads to an increase of autoantibody-secreting plasma cells in *Prdm1*^ihCd2/+ mice.

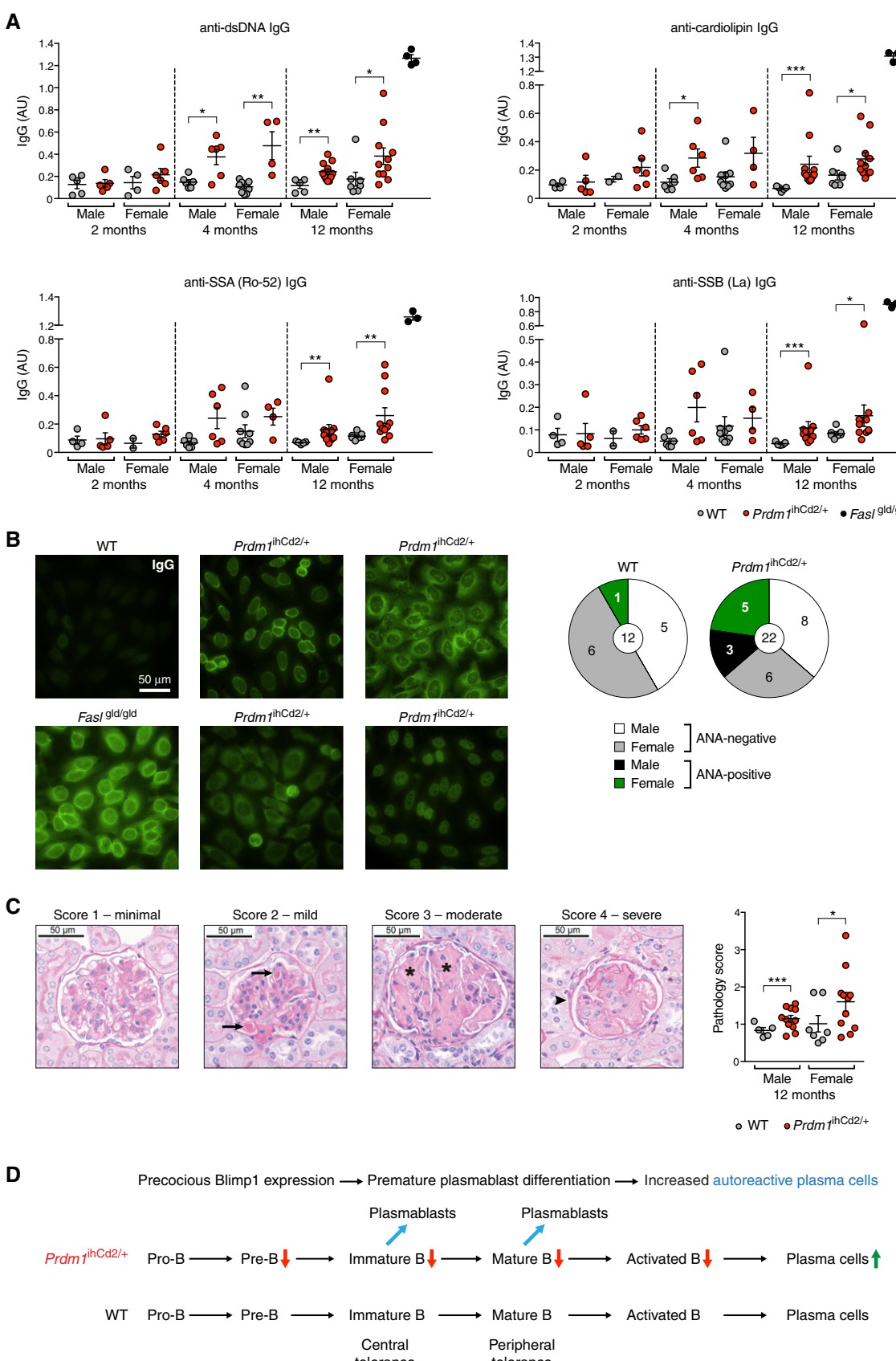

Figure 7.

level and was transcriptionally active in early B cell development long before the onset of plasmablast differentiation. Nascent *Prdm1* transcripts were, however, present at a 100-fold lower level in wild-type pre-B cells compared to the fully activated state in wild-type plasmablasts (Fig 2C). Despite transcriptional activity, mature *Prdm1* mRNA did not accumulate in early B cell development possibly due to stringent posttranscriptional control. Low expression of *Blimp1* mRNA was reported only for peritoneal B-1 and splenic MZ B cells (Fairfax *et al*, 2007), suggesting that the posttranscriptional control mechanism may operate less stringently in these two B cell types. Posttranscriptional regulation is often mediated by micro-RNAs or RNA-binding proteins that interact with microRNA targets or AU-rich elements in the 3′ UTR of mRNAs, thereby inhibiting translation and/or inducing mRNA decay (Pasquinelli, 2012; Turner *et al*, 2014). Although two microRNAs (miR-30b,d,e and miR-125b) and the RNA-binding protein ZFP36L1 have been implicated in the posttranscriptional control of *Prdm1* mRNA (Gururajan *et al*, 2010; Nasir *et al*, 2012; Parlato *et al*, 2013; Kassambara *et al*, 2017), *Prdm1* mRNA and Blimp1 protein expression did not increase upon deletion of 90% of the *Prdm1* 3′ UTR sequences in developing B cells of *Prdm1*$^{\Delta 3'U(90)/\Delta 3'U(90)}$ mice. These data suggest that micro-RNAs and RNA-binding proteins are either not involved in the posttranscriptional control of *Prdm1* mRNA or mediate their effect through the residual consensus AU-rich element and microRNA-binding site, which are still present in the *Prdm1*$^{\Delta 3'U(90)}$ allele.

By inserting the MoMLV enhancer between the stop codon and 3′ UTR of the *Prdm1*$^{\text{ihCd2}}$ allele, we created an ideal mouse model for studying the effect of ectopic Blimp1 expression in B cells, as Blimp1 expression was high enough to observe a B cell developmental defect in heterozygous *Prdm1*$^{\text{ihCd2/+}}$ mice, while a 2-fold higher level of Blimp1 protein already eliminated all B cells in homozygous *Prdm1*$^{\text{ihCd2/ihCd2}}$ mice. B cell subsets from the pre-B cell stage onwards were reduced in *Prdm1*$^{\text{ihCd2/+}}$ mice largely due to increased apoptosis. This finding is consistent with a previous report demonstrating that ectopic Blimp1 expression in established immature and mature B cell lines activates a strong apoptotic response (Messika *et al*, 1998). The observed cell death is likely caused by the strong interference of Blimp1 with the normal B cell gene expression program by activating and repressing many genes in pro-B and pre-B cells of *Prdm1*$^{\text{ihCd2/+}}$ mice. Unexpectedly, Blimp1 bound only to a small fraction of the Blimp1-regulated genes in pro-B and pre-B cells, suggesting that Blimp1 predominantly regulated gene expression in early B cells in an indirect manner in marked contrast to its significantly higher occupancy at regulated genes in terminally differentiated plasmablasts (Minnich *et al*, 2016). Blimp1 deregulated the expression of many transcription factors including those encoded by the directly repressed target genes *Spib*, *Hhex*, *Aff3*, *Irf2bp2*, and *Pax5*, which together likely mediate the indirect effects of Blimp1 in *Prdm1*$^{\text{ihCd2/+}}$ B cells. In addition to transcription factors, Blimp1 deregulated the expression of multiple cell surface receptors and intracellular signal transducers, suggesting that ectopic Blimp1 interfered with normal signaling in B cells.

GC B cells were strongly reduced in *Prdm1*$^{\text{ihCd2/+}}$ mice, although the few residual GC B cells did not express hCD2 (Blimp1) and showed normal expression of the essential regulator Bcl6. T$_{\text{FH}}$ cells, which provide T cell help for GC B cell development (Vinuesa *et al*, 2016), were also reduced in immunized

*Prdm1*$^{\text{ihCd2/+}}$ mice. However, these T$_{\text{FH}}$ cells expressed hCD2 (Blimp1) and exhibited low Bcl6 expression, as Blimp1 is known to repress *Bcl6* in T$_{\text{FH}}$ cells (Johnston *et al*, 2009). Consequently, the loss of GC B cells in *Prdm1*$^{\text{ihCd2/+}}$ mice was not a B cell-intrinsic phenotype, as the presence of "wild-type" T$_{\text{FH}}$ cells efficiently supported the differentiation of *Prdm1*$^{\text{ihCd2/+}}$ GC B cells in mixed bone marrow chimeras.

A prominent B cell-intrinsic feature of ectopic Blimp1 expression was the strong increase in plasma cells of extrafollicular origin, which was in stark contrast to the reduced B cell development and almost complete absence of GC B cells in *Prdm1*$^{\text{ihCd2/+}}$ mice. Notably, immature and mature B cells of the *Prdm1*$^{\text{ihCd2/+}}$ genotype rapidly differentiated *in vitro* to plasmablasts in response to TLR signaling, which strongly suggests that precocious Blimp1 expression in *Prdm1*$^{\text{ihCd2/+}}$ B cells led to increased plasma cell numbers by promoting premature plasmablast differentiation of B cells (Fig 7D). In this context, it is interesting to note that the lowly Blimp1-expressing B-1 and MZ B cells of wild-type mice also undergo efficient plasmablast differentiation in response to TLR signaling (Fairfax *et al*, 2007; Genestier *et al*, 2007). Hence, the low Blimp1 expression may also contribute to the enhanced plasmablast differentiation of these wild-type B cell subsets (Fairfax *et al*, 2007). As a likely consequence of increased plasma cell generation, *Prdm1*$^{\text{ihCd2/+}}$ mice developed, with progressing age, an autoimmune disease, which was characterized by the appearance of autoantibodies and a moderate form of glomerulonephritis. Based on the autoimmune phenotype of this mouse model, we hypothesize that human *PRDM1* mutations, which may result in premature Blimp1 expression in immature and mature B cells, could cause a predisposition for the development of autoimmune diseases such as SLE and RA. This predisposition could be achieved by increasing the *PRDM1* transcription rate in B cells through mutation of its regulatory elements, by stabilizing the *PRDM1* mRNA through inactivation of its posttranscriptional regulation or by stabilizing the Blimp1 protein by mutations that prevent polyubiquitination and subsequent degradation of Blimp1 (Yang *et al*, 2014).

The majority (55–75%) of early immature B cells express autoreactive BCRs as a consequence of the vast antibody diversity generated by V(D)J recombination (Wardemann *et al*, 2003). A large fraction of the immature B cells with autoreactive BCRs are eliminated in the bone marrow by central tolerance mechanisms involving receptor editing, apoptotic deletion, or AID-mediated elimination (Nemazee, 2017; Cantaert *et al*, 2015; Fig 7D). Although Blimp1 represses *Aicda* (AID) transcription in plasma cells (Minnich *et al*, 2016), ectopically expressed Blimp1 in immature B cells did not repress *Aicda* expression (data not shown), suggesting that the AID-mediated elimination of autoreactive immature B cells (Cantaert *et al*, 2015) was not affected in *Prdm1*$^{\text{ihCd2/+}}$ mice. Self-reactive immature B cells in the bone marrow are, however, known to be responsive to CpG stimulation and thus have the potential to circumvent negative selection by prematurely differentiating to plasmablasts that secrete autoantibodies (Azulay-Debby *et al*, 2007). While peripheral tolerance silences mature B cells with autoreactive BCRs through anergy induction (Theofilopoulos *et al*, 2017), it is conceivable that precocious Blimp1 expression may promote TLR-mediated differentiation of anergic mature B cells to autoreactive plasma cells in

*Prdm1*$^{ihCd2/+}$ mice (Fig 7D). Ectopic Blimp1 expression in immature and mature B cells can enhance premature plasma cell differentiation in two ways (Fig 7D). First, the enhanced apoptosis of Blimp1-expressing B cells likely results in an increase of cellular debris, apoptotic blebs, and extruded nuclei, which exposes self-antigens to polyreactive BCRs that, upon endocytosis, present these self-antigens to endosomal TLRs (TLR3,7,8,9), thus resulting in TLR activation. Second, ectopic Blimp1 expression partially activates the plasma cell program in developing B cells by interfering with the intrinsic B cell gene expression pattern and by regulating plasmablast-specific genes, which may further accelerate plasmablast differentiation, thus allowing autoreactive immature B cells to evade central tolerance and anergic B cells to escape anergy control by differentiating to plasma cells. In summary, the *Prdm1*$^{ihCd2/+}$ mouse model of ectopic Blimp1 expression has identified a novel mechanism that can explain how Blimp1 as a risk factor contributes to the development of autoimmune disease.

# Materials and Methods

Detailed methods can be found in the Appendix Supplementary Methods available online.

### Mice

The following mice were maintained on the C57BL/6 genetic background: *Prdm1*$^{ihCd2/ihCd2}$ (Minnich *et al*, 2016), *Prdm1*$^{Gfp/+}$ (Kallies *et al*, 2004), Eβ$^{-/-}$ (Bouvier *et al*, 1996), J$_H$T (Gu *et al*, 1993), *Rosa26*$^{BirA/BirA}$ (Driegen *et al*, 2005), *Pax5*$^{iGfp/iGfp}$ (Fuxa & Busslinger, 2007), *Rag2*$^{-/-}$ (Shinkai *et al*, 1992), *Fasl*$^{gld/gld}$ (Takahashi *et al*, 1994), and transgenic *Vav-Bcl2* (Ogilvy *et al*, 1999) mice. All animal experiments were carried out according to valid project licenses, which were approved and regularly controlled by the Austrian Veterinary Authorities.

### Generation of *Prdm1*$^{\Delta 3'U(90)/\Delta 3'U(90)}$ mice

The 3′ UTR in the endogenous *Prdm1* locus was deleted by CRISPR/Cas9-mediated genome editing by co-injecting mouse zygotes with *Cas9* mRNA and two specific sgRNAs (Table EV3). PCR genotyping with the primers shown in Table EV3 yielded a 399-bp and 218-bp PCR fragment for the wild-type and the *Prdm1*$^{\Delta 3'U(90)/+}$ allele, respectively.

### Antibodies

The following monoclonal antibodies were used for flow cytometry: B220/CD45R (RA3-6B2), CD3ε (145-2C11), CD4 (GK1.5), CD5 (53-7.3), CD8α (53-6.7), CD11b/Mac1 (M1/70), CD19 (1D3), CD21 (7G6), CD22 (Cy34.1), CD23 (B3B4), CD25 (PC61), CD28 (37.51), CD40 (3/23), CD44 (IM7), CD49b (DX5), CD62L (MEL-14), CD69 (H1.2F3), CD80 (16-10A1), CD86 (GL1), CD90.2/Thy1.2 (30-H12), CD95/Fas (Jo2), CD117/c-Kit (2B8), CD127/IL7Rα (A7R34), CD135/Flt3 (A2F10), CD138 (281-2), CD279/PD-1 (J43), CXCR5 (2G8), F4/80 (CI:A3-1), GL7 (GL7), Gr1 (RB6-8C5), IgD (11-26c.2a), IgM (II/41), Ly6C (6C3), Ly6D (49-H4), MHCII (M5-114), NK1.1 (PK136), Sca-1 (D7), TCRβ (H57-597), TCRγδ (GL3), and human CD2 (RPA-2.10).

### Generation of bone marrow chimeras

For the generation of mixed bone marrow chimeras, donor-derived bone marrow cells of the indicated genotypes (Fig 6A) were stained with PE-conjugated, lineage-specific antibodies (CD19, CD4, CD8α, TCRβ, TCRγδ, NK1.1, and CD49b) followed by magnetic depletion of the PE-labeled cells using MACS cell separation (Miltenyi Biotec). The donor cells, mixed at a 1:15 ratio (J$_H$T : Eβ$^{-/-}$ or Eβ$^{-/-}$ *Prdm1*$^{ihCd2/+}$), were intravenously injected into lethally irradiated (1,000 rads) *Rag2*$^{-/-}$ recipients.

### Injection of apoptotic thymocytes

To generate apoptotic cells, thymocytes of 6- to 8-week-old C56BL/6 mice were cultured for 6 h at 22°C in IMDM medium containing 10% FCS (GE Healthcare; A15-101), as described (Duhlin *et al*, 2016). Approximately $1 \times 10^7$ apoptotic thymocytes were intravenously injected into each mouse.

### Immunization, ELISPOT, and ELISA analyses

The immune response to a T cell-dependent antigen was studied by intraperitoneal injection of 100 μg of NP-KLH (Biosearch Technologies) in alum. The frequencies of NP-specific IgM antibody-secreting cells (ASCs) were determined in the spleen by enzyme-linked immunospot (ELISPOT) assay, as described (Smith *et al*, 1997).

The serum titer of NP-specific IgM, IgG1, and IgG2b antibodies was determined by enzyme-linked immunosorbent assay (ELISA; Smith *et al*, 1997) by using ELISA plates (Sigma-Aldrich), which were coated with 25 μg/ml of NP$_7$-BSA or NP$_{24}$-BSA to capture high-affinity IgG1 or total NP-specific IgM, IgG1, and IgG2b antibodies, respectively.

### ELISA measurements of autoantibodies

ELISA plates coated with mouse liver DNA were incubated with mouse serum for 2 h at 22°C. Anti-DNA-specific antibodies were detected by incubation with horseradish peroxidase-coupled goat anti-mouse IgG or goat anti-mouse IgM antibodies (SouthernBiotech) in the presence of the TMB substrate (Biolegend). The absorbance was measured at 650 nm in an Epoch Microplate Spectrophotometer (BioTek Instruments). For measuring anti-cardiolipin antibodies, ELISA plates were coated with cardiolipin (Sigma-Aldrich) overnight. Serum was added after blocking, and antigen-reactive IgG and IgM were measured with alkaline phosphate- or horseradish peroxidase-conjugated anti-mouse antibodies (SouthernBiotech). Antibodies against SSA (Ro-52) and SSB (La) were measured using commercial kits (all from Signosis Inc) following the manufacturer's instruction.

### Indirect immunofluorescence assay using HEp-2 slides

Diluted mouse serum (1:100 in PBS) was incubated on HEp-2 slides (Orgentech) for 30 min at 22°C, before the slides were washed twice for 5 min with PBS. For detection of mouse IgM or IgG, the slides were incubated for 30 min at 22°C with an Alexa488-conjugated goat anti-mouse IgM antibody or an Alexa488-conjugated goat anti-mouse IgG (H + L) antibody (both from Thermo Fisher Scientific)

as a secondary antibody. Following two washing steps, DAPI-containing mounting medium (Life Technologies) was added together with a cover slip. Images were acquired with a Zeiss Axio Imager 2 microscope and were analyzed with the Fiji software.

### ChIP-seq analysis of Blimp1 binding

Chromatin of $1 \times 10^8$ *in vitro* cultured pro-B cells from *Prdm1*<sup>ihCd2/+</sup> mice was prepared using a lysis buffer containing 0.25% SDS and was then subjected to ChIP with anti-V5 agarose beads (Sigma-Aldrich), as described (Wöhner *et al*, 2016). About 1–5 ng of ChIP-precipitated DNA was used for library preparation and Illumina deep sequencing (Table EV4).

### RNA-sequencing

RNA from *ex vivo* sorted B cells was isolated with the RNeasy Plus Mini Kit (Qiagen). mRNA was obtained by two rounds of poly(A) selection and used for library preparation and Illumina deep sequencing as described (Minnich *et al*, 2016).

### Bioinformatic analysis of RNA- and ChIP-seq data

The bioinformatic analysis of RNA- and ChIP-seq data was performed as described in detail (Minnich *et al*, 2016).

### Statistical analysis

Statistical analysis was performed with the GraphPad Prism 7 software. The two-tailed Student's *t*-test analysis was used to assess the statistical significance of differences between two experimental groups except for ELISA data, which were analyzed using the Mann–Whitney *U*-test.

### Accession numbers

RNA-seq, ChIP-seq, and GRO-seq data (Table EV4), which are first reported in this study, are available at the Gene Expression Omnibus (GEO) repository under the accession numbers GSE111692. Previously published ATAC-seq, ChIP-seq, and RNA-seq datasets are available under the GEO accession numbers indicated in Table EV4.

**Expanded View** for this article is available online.

## Acknowledgements
We thank G. Schmauß and M. Weninger for FACS sorting, A. Sommer's team at the Vienna Biocenter Support Facilities GmbH (VBCF) for Illumina sequencing, and T. Engelmeier at VBCF for histological service. This research was supported by Boehringer Ingelheim, the European Community's Seventh Framework Program (European Research Council Advanced Grant 291740-LymphoControl), and the Austrian Industrial Research Promotion Agency (Headquarter Grant FFG-852936).

## Author contributions
PB performed most experiments; MW performed the ELISPOT, ELISA, BrdU incorporation, and cell injection experiments; MM generated the *Prdm1*<sup>ihCd2</sup> allele and discovered the Blimp1 overexpression phenotype of the *Prdm1*<sup>ihCd2/+</sup> mouse; HT generated the GRO-seq data; MG measured the anti-cardiolipin, anti-SSA, and anti-SSB antibody titers; MCIK provided advice for analysis of the autoimmune phenotype; AK evaluated the pathology of the kidneys; MF and MJ performed the bioinformatic analysis of all RNA-seq and ChIP-seq data, respectively; MB and PB planned the project and wrote the manuscript.

## Conflict of interest
The authors declare that they have no conflict of interest.

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
