## [Review Process File · The EMBO Journal]

Precocious expression of Blimp1 in B cells causes autoimmune disease with increased self-reactive plasma cells

Peter Bönelt, Miriam Wöhner, Martina Minnich, Hiromi Tagoh, Maria Fischer, Markus Jaritz, Anoop Kavirayani, Manasa Garimella, Mikael C. I. Karlsson and Meinrad Busslinger

Review timeline:

Submission date:	8th Jun 2018
Editorial Decision:	23rd Jul 2018
Revision received:	27th Sep 2018
Editorial Decision:	22nd Oct 2018
Revision received:	29th Oct 2018
Accepted:	2nd Nov 2018

Editor: Karin Dumstrei

Transaction Report:

1st Editorial Decision

23rd Jul 2018

Thank you for submitting your study to The EMBO Journal. Your study has now been seen by three referees and their comments are provide below.

As you can see, the referees find the analysis interesting and support publication here. They raise a number of different points that should be straightforward enough to address. I would therefore like to invite you to submit a suitably revised manuscript that takes the referee comments into consideration.

REFEREE REPORTS:

Referee #1:

Pdrm1/Blimp1 has been identified as a risk factor for autoimmunity. Based on their observation that the Pdrm1 locus is partially activated during early B cell development in mice, the authors assess the consequences of premature Blimp1 protein expression throughout B cell development in a novel mouse model. The authors provide a comprehensive and thorough analysis of the mice. They show that heterozygous Blimp1 expression in Pdrm1^{ih}Cd2^{+/+} mice blocks B cell development at the pro-B to pre-B transition and leads to increased development of antibody secreting cells. Transfer experiments demonstrate that this is a B-cell intrinsic effect. Aged Pdrm1^{ih}Cd2^{+/+} mice develop autoantibody titers and glomerulonephritis suggesting that ectopic Blimp1 expression is a risk factor for autoimmunity. The manuscript is well written and the authors are careful in their conclusions. In

summary, this is a very solid assessment of the consequences of premature Blimp1 in early B cell development and demonstrates its relevance as a risk factor for autoimmunity. The findings are novel, relevant, and unexpected.

However, a few points need to be clarified. The authors state in the introduction that they „hypothesized that PRDM1 mutations in SLE or RA patients may lead to deregulated expression of the Blimp1 protein in the B cell lineage by overriding the posttranscriptional control mechanism" and "tested this hypothesis with a mouse model". I do not understand how the *Pdrml1hCd2/+* mouse models the SNPS in the intergenic region between the PRDM1 and ATG5 locus that have been identified as risk factors in SLE and RA. Along this line, I find it difficult to connect the data in figure 1 with the development of the *Pdrml1hCd2/+* mouse model. The authors could consider moving Figure 1 into the supplement or presenting it as the last figure of the manuscript. The justification for the generation and analysis of the mouse model is given by the assumption that premature Blimp1 expression may lead to autoimmunity. I do not see a need to present this in a highly speculative context with SLE and RA in the results section rather than the discussion.

Another point that is unclear to me relates to the GC response and the development of autoantibodies. *Pdrml1hCd2/+* mice develop autoimmunity with age. Are the autoantibodies really the product of immature B cells that failed to be regulated by tolerance mechanisms in the bone marrow (or periphery)? In other words, can the authors exclude that the autoreactive antibody-secreting cells that accumulate with age (especially if they produce IgG) originate from (rare and potentially expanded) PCs that developed in GC responses? If the autoreactive PCs in *Pdrml1hCd2/+* mice develop from immature B cells, why are the IgG titers high? The NP-KLH response is dominated by IgM.

I find these points important to clarify given the fact that autoantibodies in SLE and RA are typically affinity matured. At least I am not aware of evidence that deregulated Blimp1 expression in human SLE/RA is associated with a complete loss of Tfh cells and GCs or that the autoantibody producing cells in SLE/RA originate directly from immature cells.

Additional points:

Do the *Pdrml1hCd2/+* mice show differences in overall IgM and IgG serum titers and IgG subclasses compared to wt mice?

The authors show that GC B cells fail to develop in *Pdrml1hCd2/+* mice, as a consequence of ectopic Blimp1 expression in T cells and reduced Tfh cell numbers. In contrast, GC B cell numbers were not significantly lower in bone marrow chimeras with wt T cells and their frequency was even increased. Could (some of) these cells be GC-derived?

Did the authors assess the autoantibody response in the BM chimeric mice? One would expect to see even higher titers of autoantibodies in these mice with stronger GC responses.

Referee #2:

The manuscript from Bönelt and colleagues analyzes how premature Blimp1 expression alters early B cell development and favors the development of plasma cells that may produce autoantibodies. While some of the data may be expected since early expression of Blimp1 represses Pax5 essential for B cell commitment and development and therefore decreases B cell production, ATAC-seq, ChIP-seq and RNA-seq analyses described in the manuscript characterize in depth the molecular events leading to this phenotype. In addition and less expected is the development of an autoimmune phenotype associated with early Blimp1 expression. The B cell tolerance mechanisms affected by early Blimp1 expression are less clear and not as well investigated in the manuscript. It has been reported that AID plays an important role in the establishment of central B cell tolerance (several reports from the Meffre and Kelsoe teams). Since Blimp1 represses AID expression, one may expect that developing autoreactive B cells that express Blimp1 in the bone marrow may fail to be removed. The authors may test this hypothesis or at least discuss it.

In addition, the authors should soften the relevance of their observation to GWAS identified PRDM1 variants associated with SLE and RA in that their overexpression of Blimp1 may be well superior to what these gene polymorphisms may induce in patients.

Altogether, this manuscript is a well-designed and informative study on the regulation of early B cell development that is altered by the precocious expression of Blimp1, leading to autoimmunity.

Referee #3:

This report essentially describes the consequences on B cell development and differentiation of constitutive over-expression of the gene *Prdm1*, encoding *Blimp1*. Essentially the authors report first that there is evidence for *Prdm1* transcription in early stages of B cell development, despite there being little full transcript detected. While they show this post-transcriptional regulation is not due to a series of potential and real regulatory regions in the 3'UTR, they don't define the mechanism beyond this. They do show, however, that enforced expression driven by a heterologous enhancer, can overcome this regulation and result in significant accumulation of *Prdm1* mRNA and presumably protein. This has the effect of reducing B cell development from the Pro-B stage onwards, a reduction to which pre-B cell apoptosis contributes. Perhaps curiously, the means of this diminution is not fully elucidated except for identifying a modest footprint for *Blimp1* binding to the genome of the transgenic Pro-B cells, and a similar modest impact on gene expression, both activation and repression. Despite this somewhat unsatisfactory state of affairs at the end of B cell development, in the periphery there were significant and perhaps more expected outcomes. Antibody secreting cell (ASC) frequency was increased in bone and spleen unmanipulated mice and IgM ASC in mice challenged with a T-cell dependent antigen, although IgG output was reduced. GC B cell and Tfh frequency were reduced. In an unexpected result, chimeric mice constructed to limit *Prdm1* overexpression to only the B cell lineage recapitulated the deficit in B-cell production in bone marrow and spleen, but showed effectively normal numbers of ASC in spleen and bone marrow and normal GC frequency in spleen, albeit in unimmunized mice, suggesting the over-expression of *Prdm1* in CD4 T cells was sufficient to divert them from a Tfh fate. Finally, the authors show that the transgenic mice develop a mild to moderate antibody mediated autoimmune disease, characterised by the development of glomerulonephritis and the deposition of immune complexes in the kidneys, a disease development that can be exacerbated by multiple injections of apoptotic cells. Overall it is very interesting and could make a significant contribution to understanding the constraints on B cell differentiation and their relationship to the development of autoimmune disease.

In general this is a very interesting study and one that is quite meticulous in its attention to detail, with some exceptions. Some of my issues are related to its structure, some with interpretation and some with minor details.

1. I am not convinced by the argument for the study. It is implied (at least I took this implication) that this mouse model is connected in some way with the *Prdm1*-linked polymorphisms reported for SLE patients. There are no data presented or quoted that indicate whether the human alleles are associated with increased *Prdm1* expression during development. This should be clarified.
2. The failure to identify a mechanism for the B cell deficit is frustrating - as it must be for the authors also. Are there any kinetic data to suggest why despite lower production, peripheral numbers don't accumulate? Are the increased PC immature and/or turning over more rapidly? If yes, please show these; if no, please indicate this as a possibility.
3. The production of auto-antibodies in mice is often associated with altered plasma cell compartments (eg *Lyn*-deficient; *Ets1*-deficient). Could the authors provide an indication if the autoantibody titres in the transgenic mice are significantly increased relative to the serum immunoglobulin amounts? That is, if one normalises the autoantibodies for total serum Ig, are they still significantly increased? This speaks to mechanism.
4. What are the immunoglobulin titres in the unmanipulated transgenic mice? At 4 months for example?
5. What are the number of splenocytes in the transgenic mice? It is challenging to interpret cell frequency data (eg 5C) when total numbers are not available (apologies if I have missed these).
6. What are the isotypes of the expanded ASC in the spleens and BM of the unmanipulated transgenic mice? This is related to (4)
7. The ability to compare the results in Fig 5 and 6 is somewhat inhibited as the mice in 6 were not challenged with antigen. Are there any data from such experiments? Are there measurements for Tfh frequency in the chimeras in Fig 6?
8. Could the authors explain "TPM", as "mean expression value" is probably not sufficient for most readers.
9. In Fig 4, could the authors explain that Pre-PB are *Blimp1* deficient? It may not be obvious to most readers.

10. In Fig 5 legend, correct (back) to (black).
8-10 are minor.

1st Revision - authors' response

27th Sep 2018

Point-by-point reply to the reviewer's comments

Referee #1:

Prdm1/Blimp1 has been identified as a risk factor for autoimmunity. Based on their observation that the Prdm1 locus is partially activated during early B cell development in mice, the authors assess the consequences of premature Blimp1 protein expression throughout B cell development in a novel mouse model. The authors provide a comprehensive and thorough analysis of the mice. They show that heterozygous Blimp1 expression in Prdm1^{ihCd2/+} mice blocks B cell development at the pro-B to pre-B transition and leads to increased development of antibody secreting cells. Transfer experiments demonstrate that this is a B-cell intrinsic effect. Aged Prdm1^{ihCd2/+} mice develop autoantibody titers and glomerulonephritis suggesting that ectopic Blimp1 expression is a risk factor for autoimmunity. The manuscript is well written and the authors are careful in their conclusions. In summary, this is a very solid assessment of the consequences of premature Blimp1 in early B cell development and demonstrates its relevance as a risk factor for autoimmunity. The findings are novel, relevant, and unexpected.

We thank the reviewer for his/her positive assessment of our manuscript.

Point 1: However, a few points need to be clarified. The authors state in the introduction that they „hypothesized that PRDM1 mutations in SLE or RA patients may lead to deregulated expression of the Blimp1 protein in the B cell lineage by overriding the posttranscriptional control mechanism" and "tested this hypothesis with a mouse model". I do not understand how the Prdm1^{ihCd2/+} mouse models the SNPs in the intergenic region between the PRDM1 and ATG5 locus that have been identified as risk factors in SLE and RA.

As suggested by the reviewer, we eliminated the corresponding statements at the end of the first paragraph (page 3, top) and at the start of the last paragraph (page 4, bottom) of the introduction as well as at the beginning of the paragraph "Premature expression of Blimp1..." (page 7, top) in the result section. We will thus no longer mention the intergenic SNPs in the context of our mouse model in the introduction and result section, as requested.

Point 2: Along this line, I find it difficult to connect the data in figure 1 with the development of the Prdm1^{ihCd2/+} mouse model. The authors could consider moving Figure 1 into the supplement or presenting it as the last figure of the manuscript. The justification for the generation and analysis of the mouse model is given by the assumption that premature Blimp1 expression may lead to autoimmunity. I do not see a need to present this in a highly speculative context with SLE and RA in the results section rather than the discussion.

As suggested by the reviewer, we have eliminated the relevant sentences dealing with the speculative hypothesis and justification for the Blimp1 overexpression mouse model in the introduction and result section (see point 1 above) and will mention the implication of the autoimmune phenotype of the Prdm1^{(ihCd2/+)} mouse model for human autoimmune disease only in the discussion section. However, we disagree with the reviewer with regard to moving Figure 1 to a supplementary Figure. Our finding that the Prdm1 locus is transcriptionally active already in early B cell development is totally unexpected and highly interesting. Moreover, the other two reviewers did not criticize this point, while reviewer #3 even highlighted this finding in his/her review.

Point 3: Another point that is unclear to me relates to the GC response and the development of

autoantibodies. *Prdm1ihCd2/+* mice develop autoimmunity with age. Are the autoantibodies really the product of immature B cells that failed to be regulated by tolerance mechanisms in the bone marrow (or periphery)? In other words, can the authors exclude that the autoreactive antibody-secreting cells that accumulate with age (especially if they produce IgG) originate from (rare and potentially expanded) PCs that developed in GC responses? If the autoreactive PCs in *Prdm1ihCd2/+* mice develop from immature B cells, why are the IgG titers high? The NP-KLH response is dominated by IgM.

We have shown that FO B cells of the spleen, in addition to immature B cells, can also undergo accelerated plasmablast differentiation in response to CpG, LPS or IL-4, IL-5 plus anti-CD40 stimulation (Figure S5H-K). We have also highlighted this fact in the summary Figure 7D of the previously submitted manuscript. It is therefore conceivable that anergic autoreactive FO B cells in peripheral lymphoid organs can also undergo premature differentiation to plasma cells, which is further supported by the presence of autoreactive IgG in the serum *Prdm1ihCd2/+* mice (Figure 7A). We now mention this issue more explicitly in the discussion of the revised manuscript (on page 23). Class switch recombination to IgG isotypes is thought to occur upon B-T cell interaction before the decision is made whether an activated B cell will enter the GC B cell pathway or undergo extrafollicular differentiation to plasmablasts. Hence, the presence of IgG antibodies does not indicate whether the corresponding plasma cells have been generated by the extrafollicular or GC B cell pathway. Two arguments suggest that the autoreactive IgG antibodies detected in *Prdm1ihCd2/+* mice do not originate from GC B cells. First, GC B cells are strongly reduced in *Prdm1ihCd2/+* mice in steady state (Figure S5G) or upon NP-KLH-immunization (Figure 5E). Second, the frequency of somatic hypermutation was lower in plasma cells from the spleen of non-immunized *Prdm1ihCd2/+* mice compared to wild-type mice (shown in Figure S5F), which is expected, if plasma cells are primarily generated by the extrafollicular pathway.

Point 4: I find these points important to clarify given the fact that autoantibodies in SLE and RA are typically affinity matured. At least I am not aware of evidence that deregulated Blimp1 expression in human SLE/RA is associated with a complete loss of Tfh cells and GCs or that the autoantibody producing cells in SLE/RA originate directly from immature cells.

This general statement of the reviewer is correct. However, we are not aware of evidence that SLE patients with SNPs in the *PRDM1* locus have been specifically investigated with regard to the presence of GC B cells, T_{FH} cells or antibody affinity maturation. It thus remains a possibility that these SLE patients differ from other SLE patients in this regard. Moreover, AID-deficient patients in the absence of affinity maturation have a high frequency of anti-nuclear IgM antibodies in their serum and are prone to develop autoimmune disease (Meyers et al., 2011, PNAS 108, 11554-59; Kuraoka et al., 2011, PNAS 108, 11560-65; Cantaert et al., 884-95).

Additional points:

Point 5: Do the *Prdm1ihCd2/+* mice show differences in overall IgM and IgG serum titers and IgG subclasses compared to wt mice?

We measured the total IgM and IgG titers in the serum of wild-type and *Prdm1ihCd2/+* mice at the age of 2, 4 and 12 months by ELISA (new Figure S5B) and used the ELISPOT assay to determine the number of plasma cells that secreted IgG1, IgG2b and IgG3 antibodies in the spleen and bone marrow of *Prdm1ihCd2/+* and wild-type mice (new Figure S5C). These data indicate that the serum concentration of total IgM and IgG are increased at all three time points in *Prdm1ihCd2/+* mice compared to wild-type littermates. Likewise, the IgG isotype-secreting plasma cells were also increased in *Prdm1ihCd2/+* mice consistent with the general increase of total plasma cells in these mice. We now mention these data on pages 13 and 14 in the result section.

Point 6: The authors show that GC B cells fail to develop in *Prdm1ihCd2/+* mice, as a consequence of ectopic Blimp1 expression in T cells and reduced Tfh cell numbers. In contrast, GC B cell numbers were not significantly lower in bone marrow chimeras with wt T cells and their frequency was even

increased. Could (some of) these cells be GC-derived? Did the authors assess the autoantibody response in the BM chimeric mice? One would expect to see even higher titers of autoantibodies in these mice with stronger GC responses.

This reviewer and reviewer #3 (point 7) raise an interesting question. Unfortunately, we did not measure the frequency of T_{FH} cells or the titer of autoreactive antibodies in the chimeric mice analyzed in Figure 6. Moreover, we lost the experimental *Prdm1*(ihCd2/+) *Eb*(-/-) mice at the time of submission and, hence, these mice were no longer available for generating chimeric mice to perform the immunization experiments requested by the reviewers. Regenerating the *Prdm1*(ihCd2/+) *Eb*(-/-) genotype would take minimally 4 months, before we could perform the bone marrow transfer experiment. Thereafter, we would have to wait for another 4 months until we could reliably measure the titer of autoantibodies (see Figure 7). Hence 8 months would be the earliest time point, at which we could obtain data for the experiments suggested by reviewers #1 and #3. As the timeframe for these experiments is unreasonably long, we could not provide an answer to the question raised by both reviewers.

Referee #2:

The manuscript from Bönelt and colleagues analyzes how premature Blimp1 expression alters early B cell development and favors the development of plasma cells that may produce autoantibodies. While some of the data may be expected since early expression of Blimp1 represses Pax5 essential for B cell commitment and development and therefore decreases B cell production, ATAC-seq, ChIP-seq and RNA-seq analyses described in the manuscript characterize in depth the molecular events leading to this phenotype. In addition and less expected is the development of an autoimmune phenotype associated with early Blimp1 expression.

Point 1: The B cell tolerance mechanisms affected by early Blimp1 expression are less clear and not as well investigated in the manuscript. It has been reported that AID plays an important role in the establishment of central B cell tolerance (several reports from the Meffre and Kelsoe teams). Since Blimp1 represses AID expression, one may expect that developing autoreactive B cells that express Blimp1 in the bone marrow may fail to be removed. The authors may test this hypothesis or at least discuss it.

We performed the requested experiment using the conditions described by Kuraoka et al. (Cell Rep. 18, 1627-35). Immature B cells (CD19⁺B220⁺IgM⁺IgD⁻) were isolated by flow cytometry from the bone marrow of *Prdm1*(ihCd2/+) and wild-type mice and were then stimulated with CpG oligodeoxynucleotides (ODN 1826; 0.2 mM) and anti-IgM antibody (M41.42; 3 mg/ml) in IMDM medium (supplemented with 10% FCS) for 24 h prior to RNA preparation and RT-qPCR analysis. As shown in Figure 1 (for reviewer), the *Aicda* mRNA was expressed at similar levels in immature B cells of the *Prdm1*(ihCd2/+) genotype compared to wild-type immature B cells before treatment as well as after stimulation. Moreover, we could not detect Blimp1 binding in the first intron of the *Aicda* gene in pro-B cells by ChIP-seq analysis in marked contrast to plasmablasts (Minnich et al., Nat. Immunol. 17, 331-43). Hence, these data suggest that the autoimmunity observed in *Prdm1*(ihCd2/+) mice is not caused by loss of AID through Blimp1-mediated repression of *Aicda* transcription in immature *Prdm1*(ihCd2/+) B cells. We now mention this result in the discussion section (page 23, middle).

Figure 1 (for reviewer). (A) Similar expression of *Aicda* mRNA (normalized to *Thp* mRNA) in untreated and CpG- plus anti-IgM-stimulated immature B cells from the bone marrow of *Prdm1*(ihCd2/+) and wild-type mice, as determined by RT-qPCR analysis. (A) No Blimp1 binding in the first intron of the *Aicda* gene in *Prdm1*(ihCd2/+) pro-B cells in contrast to plasmablasts, as analyzed by ChIP-seq.

Point 2: In addition, the authors should soften the relevance of their observation to GWAS identified PRDM1 variants associated with SLE and RA in that their overexpression of Blimp1 may be well superior to what these gene polymorphisms may induce in patients.

We followed the suggestion of this reviewer (see response to points 1 + 2 of reviewer #1).

Altogether, this manuscript is a well-designed and informative study on the regulation of early B cell development that is altered by the precocious expression of Blimp1, leading to autoimmunity.

We thank the reviewer for his/her positive assessment of our manuscript.

Referee #3:

This report essentially describes the consequences on B cell development and differentiation of constitutive over-expression of the gene *Prdm1*, encoding Blimp1. Essentially the authors report first that there is evidence for *Prdm1* transcription in early stages of B cell development, despite there being little full transcript detected. While they show this post-transcriptional regulation is not due to a series of potential and real regulatory regions in the 3'UTR, they don't define the mechanism beyond this. They do show, however, that enforced expression driven by a heterologous enhancer, can overcome this regulation and result in significant accumulation of *Prdm1* mRNA and presumably protein. This has the effect of reducing B cell development from the pro-B stage onwards, a reduction to which pre-B cell apoptosis contributes. Perhaps curiously, the means of this diminution is not fully elucidated except for identifying a modest footprint for Blimp1 binding to the genome of the transgenic pro-B cells, and a similar modest impact on gene expression, both activation and repression. Despite this somewhat unsatisfactory state of affairs at the end of B cell development, in the periphery there were significant and perhaps more expected outcomes. Antibody secreting cell (ASC) frequency was increased in bone and spleen of unmanipulated mice and IgM ASC in mice challenged with a T-cell dependent antigen, although IgG output was reduced. GC B cell and Tfh frequency were reduced. In an unexpected result, chimeric mice constructed to limit *Prdm1* overexpression to only the B cell lineage recapitulated the deficit in B-cell production in bone marrow and spleen, but showed effectively normal numbers of ASC in spleen and bone marrow and normal GC frequency in spleen, albeit in unimmunized mice, suggesting the over-expression of *Prdm1* in CD4 T cells was sufficient to divert them from a Tfh fate. Finally, the authors show that the transgenic mice develop a mild to moderate antibody mediated autoimmune disease, characterised by the development of glomerulonephritis and the deposition of immune complexes in the kidneys, a disease development that can be exacerbated by multiple injections of apoptotic cells. Overall it is very interesting and could make a significant contribution to understanding the constraints on B cell differentiation and their relationship to the development of autoimmune disease.

We thank the reviewer for his/her positive assessment of our manuscript.

In general this is a very interesting study and one that is quite meticulous in its attention to detail, with some exceptions. Some of my issues are related to its structure, some with interpretation and some with minor details.

Point 1: I am not convinced by the argument for the study. It is implied (at least I took this implication) that this mouse model is connected in some way with the *Prdm1*-linked polymorphisms reported for SLE patients. There are no data presented or quoted that indicate whether the human alleles are associated with increased *Prdm1* expression during development. This should be clarified. We have eliminated all speculations (in the introduction and result section) about a potential link between the *Prdm1*-linked polymorphisms reported for SLE patients and our mouse model (see response to points 1 + 2 of reviewer #1).

Point 2: The failure to identify a mechanism for the B cell deficit is frustrating - as it must be for the authors also. Are there any kinetic data to suggest why despite lower production, peripheral numbers don't accumulate? Are the increased PC immature and/or turning over more rapidly? If yes, please show these; if no, please indicate this as a possibility.

As suggested by the reviewer, we now provide kinetic data for splenic FO B cells and plasma cells. To this end, we measured the *in vivo* lifespan of Blimp1-overexpressing FO B cells and plasma cells by continuous BrdU labeling of *Prdm1*(ihCd2/+) and wild-type mice for 10 days followed by a 15-day chase period. These experiments revealed a two-fold higher incorporation of BrdU⁺ immature B cells into the stable quiescent FO B cell pool of *Prdm1*(ihCd2/+) mice at day 10 compared to that of wild-type mice (new Figure 3E,F). Moreover, half of the BrdU⁺ FO B cells were replaced by unlabeled *Prdm1*(ihCd2/+) FO B cells during the subsequent 15-day chase period in contrast to the wild-type FO B cells (Figures 3E,F). These data therefore revealed a shortened lifespan and thus more rapid turnover of *Prdm1*(ihCd2/+) FO B cells compared to control FO B cells in the spleen. We mention these data in the result section on page 10 and in the legend of Figure 3E,F. We also added a corresponding paragraph describing BrdU labeling in the Online Method on page 5.

As shown in Figure 2 (for reviewer), analysis of the BrdU labeling in splenic plasma cells revealed that BrdU was incorporated at a 1.6-fold lower level into *Prdm1*(ihCd2/+) plasma cells compared to wild-type plasma cells at day 10, suggesting that fewer cell proliferation was required for the generation of *Prdm1*(ihCd2/+) plasma cells compared to wild-type plasma cells. This result suggests that the *Prdm1*(ihCd2/+) plasma cells may have been generated through the extrafollicular pathway compared to wild-type plasma cells, some of which may have developed via the GC B cell pathway involving rapid cell proliferation.

Figure 2 (for reviewer). (A,B) BrdU labeling of splenic plasma cells from the spleen of 3-month-old *Prdm1*(ihCd2/+) mice (red dots) and wild-type littermates (gray dots). The percentage of BrdU⁺ plasma cells was determined by flow cytometry after 10 days of continuous BrdU labeling (day 10,

white bar) or after a subsequent 15-day chase period (day 25; hatched bar) without BrdU in the drinking water (A). A diagram (below) indicates the design of the BrdU labeling and chase experiments. Flow cytometric data obtained with plasma cells from one mouse of each genotype are shown in (B). The percentage of BrdU⁺ FO B cells is shown above the indicated gates (B).

Point 3: The production of auto-antibodies in mice is often associated with altered plasma cell compartments (eg Lyn-deficient; Ets1-deficient). Could the authors provide an indication if the autoantibody titres in the transgenic mice are significantly increased relative to the serum immunoglobulin amounts? That is, if one normalises the autoantibodies for total serum Ig, are they still significantly increased? This speaks to mechanism.

We have determined the titer of total IgM and IgG antibodies in the serum of *Prdm1*(ihCd2/+) and wild-type mice at 2, 4 and 12 months by ELISA (new Figure S5B). Consistent with the observed increase in plasma cells, the IgM and IgG titers were also increased in *Prdm1*(ihCd2/+) mice relative to wild-type mice. As the increase of autoreactive IgM and IgG varies for the different autoantigens analyzed (dsDNA, cardiolipin, SSA and SSB; Figures 7A and S7A), it is not possible to determine one ratio for total and autoreactive antibodies. It is, however, obvious that autoreactive antibodies were increased at a similar or sometimes slightly higher ratio in *Prdm1*(ihCd2/+) mice relative to wild-type mice.

Point 4: What are the immunoglobulin titres in the unmanipulated transgenic mice? At 4 months for example?

We measured the total IgM and IgG titers in the serum of wild-type and *Prdm1*(ihCd2/+) mice at the age of 2, 4 and 12 months by ELISA (new Figure S5B) and used the ELISPOT assay to determine the number of plasma cells that secreted IgG1, IgG2b and IgG3 antibodies in the spleen and bone marrow of wild-type and *Prdm1*(ihCd2/+) mice (new Figure S5C). For discussion of these results, see our response to point 5 of reviewer #1.

Point 5: What are the numbers of splenocytes in the transgenic mice? It is challenging to interpret cell frequency data (eg 5C) when total numbers are not available (apologies if I have missed these).

We now display the number of total cells in the spleen and bone marrow of untreated *Prdm1*(ihCd2/+) and wild-type mice at the age of 2, 4 and 12 months (new Figure S5A) and in the spleen of both genotypes at day 14 after NP-KLH immunization (new Figure S5E). As shown by these data, the cell numbers are similar in mice of both genotypes or only minimally reduced in *Prdm1*(ihCd2/+) mice relative to wild-type littermates. Hence, the cell frequencies shown in Figure 5A-G can be directly compared.

Point 6: What are the isotypes of the expanded ASC in the spleens and BM of the unmanipulated transgenic mice? This is related to (4).

As suggested by the reviewer, we used intracellular flow cytometry to determine the frequency of immunoglobulin isotype-secreting plasma cells within the plasma cell compartment in the spleen and bone marrow of untreated *Prdm1*(ihCd2/+) and wild-type mice at the age of 7 months (Figure 3 for reviewer). These data revealed similar frequencies of plasma cells secreting individual isotypes in both organs of *Prdm1*(ihCd2/+) and wild-type mice, except for a moderate increase of IgG3- and IgM-secreting plasma cells in the spleen and a decrease of IgA-secreting plasma cells in the bone marrow of *Prdm1*(ihCd2/+) mice.

Figure 3 (for reviewer). The frequency of plasma cells ($\text{Lin}^- \text{CD138}^+ \text{CD28}^+ \text{B220}^{\text{int}/-}$), secreting individual immunoglobulin isotypes, among all plasma cells was determined by intracellular flow cytometric analysis of spleen and bone marrow cells from 7-month-old *Prdm1*^{ihCd2/+} (red dots) and wild-type (grey dots) mice. Statistical data are shown as mean value with SEM and were analyzed by the Student's *t*-test; **P* < 0.05 and ***P* < 0.01.

Point 7: The ability to compare the results in Fig 5 and 6 is somewhat inhibited as the mice in 6 were not challenged with antigen. Are there any data from such experiments? Are there measurements for Tfh frequency in the chimeras in Fig 6?

This reviewer and reviewer #1 (point 6) raise an interesting question. Unfortunately, we did not measure the frequency of T_{FH} cells or the titer of autoreactive antibodies in the chimeric mice analyzed in Figure 6. For time constraints, we could not repeat the experiments, as detailed in the response to point 6 of reviewer #1,

Point 8: Could the authors explain "TPM", as "mean expression value" is probably not sufficient for most readers.

Bioinformaticians strongly recommend using TPM values (transcripts per million; Wagner et al., 2012) rather than RPKM values for comparing gene expression between different RNA-seq experiments. We added "(transcripts per million)" at the first appearance of the term TPM on page 11 in the result section and on page 15 in the Appendix Supplementary Methods. We also added "TPM, transcripts per million (Wagner et al., 2012)" at the end of the legend of Figure 1 (page 31) and the corresponding citation in the reference list (page 30).

Point 9: In Fig 4, could the authors explain that Pre-PB are Blimp1 deficient? It may not be obvious to most readers.

We now explicitly mention the genotype of the Blimp1-deficient (*Prdm1*(Gfp/Δ), blue) pre-plasmablasts in the legend of Figure 4G (page 36).

Point 10: In Fig 5 legend, correct (back) to (black).

The correction was made in the legend of Figure 5.

Change of title:

We would like to change the title to make it more informative by better describing the main message of the paper. For this, we would like to replace "autoimmunity" with "autoimmune disease", as we have shown the presence of increased autoreactive antibodies, glomerulonephritis and the deposition of immune complexes in the kidney of *Prdm1*(ihCd2/+ mice. Moreover, the Blimp1-mediated evasion of central and peripheral tolerance mechanism is cell-intrinsic and does not involve a misguided or uncontrolled immune response, as typically seen in autoimmunity.

Thank you for submitting your revised manuscript to The EMBO Journal. Your study has now been re-reviewed by referee #1 and 3 and their comments are provided below. As you can see both referees appreciate that the analysis has been strengthened and support publication here. Referee #1 suggests moving a paragraph from the introduction to the discussion. I think it is a good suggestion.

REFeree REPORTS:

Referee #1:

The authors addressed my comments. I only suggest to remove the first paragraph on SLE and RA in humans from the beginning of the introduction. It should be part of the discussion.

Referee #3:

The manuscript is significantly improved by the extensive additional experiments and rewriting that has been undertaken. In my opinion it now makes an important contribution to our understanding of aspects of B cell development and the relationship between abnormal Blimp1 expression and normal differentiation. I think it should be accepted.

YOU MUST COMPLETE ALL CELLS WITH A PINK BACKGROUND ↓
PLEASE NOTE THAT THIS CHECKLIST WILL BE PUBLISHED ALONGSIDE YOUR PAPER

Corresponding Author Name: Meinrad Busslinger
Journal Submitted to: EMBO Journal
Manuscript Number: EMBOJ-2018-100010R